# Vegetation Evolution with Dynamic Maturity Strategy and Diverse Mutation Strategy for Solving Optimization Problems

**DOI:** 10.3390/biomimetics8060454

**Published:** 2023-09-25

**Authors:** Rui Zhong, Fei Peng, Enzhi Zhang, Jun Yu, Masaharu Munetomo

**Affiliations:** 1Graduate School of Information Science and Technology, Hokkaido University, Sapporo 060-0808, Japan; enzhi.zhang.n6@elms.hokudai.ac.jp; 2Graduate School of Science and Technology, Niigata University, Niigata 950-3198, Japan; f22c128f@mail.cc.niigata-u.ac.jp; 3Institute of Science and Technology, Niigata University, Niigata 950-3198, Japan; yujun@ie.niigata-u.ac.jp; 4Information Initiative Center, Hokkaido University, Sapporo 060-0808, Japan; munetomo@iic.hokudai.ac.jp

**Keywords:** evolutionary computation, diverse mutation strategies, dynamic maturity strategy, vegetation evolution

## Abstract

We introduce two new search strategies to further improve the performance of vegetation evolution (VEGE) for solving continuous optimization problems. Specifically, the first strategy, named the dynamic maturity strategy, allows individuals with better fitness to have a higher probability of generating more seed individuals. Here, all individuals will first become allocated to generate a fixed number of seeds, and then the remaining number of allocatable seeds will be distributed competitively according to their fitness. Since VEGE performs poorly in getting rid of local optima, we propose the diverse mutation strategy as the second search operator with several different mutation methods to increase the diversity of seed individuals. In other words, each generated seed individual will randomly choose one of the methods to mutate with a lower probability. To evaluate the performances of the two proposed strategies, we run our proposal (VEGE + two strategies), VEGE, and another seven advanced evolutionary algorithms (EAs) on the CEC2013 benchmark functions and seven popular engineering problems. Finally, we analyze the respective contributions of these two strategies to VEGE. The experimental and statistical results confirmed that our proposal can significantly accelerate convergence and improve the convergence accuracy of the conventional VEGE in most optimization problems.

## 1. Introduction

Since evolutionary computation (EC) does not depend on the characteristics of optimization problems and has the advantages of parallelism and robustness, these algorithms have been successfully applied to various real-world applications, such as drug design [1,2,3], Twitter bot detection [4,5,6], anomaly detection [7,8], and engineering [9,10,11]. With their popularity in the field of optimization, numerous new EC algorithms have been proposed. For example, invasive weed optimization (IWO) [12] was inspired by the strong survival capacity of colonizing weeds to imitate the robustness, adaptation, and randomness of colonizing weeds. The remora optimization algorithm (ROA) simulates the parasitic behavior of remora to continuously optimize the current population [13], beluga whale optimization (BWO) simulates three phases of beluga whales (i.e., exploration, exploitation, and whale fall) to find the global optimum [14], and plant competition optimization (PCO) [15] assumes each feasible solution during optimization as a plant and competes with its neighbors to realize optimization.

As one of the newest EC algorithms, vegetation evolution (VEGE) repeatedly simulates the behavior of plants in different periods to balance exploration and exploitation from a fresh perspective [16]. Due to the superior performance of the conventional VEGE compared with some classic EC algorithms, e.g., differential evolution (DE) [17], particle swarm optimization (PSO) [18], and the enhanced fireworks algorithm (EFWA) [19], it has attracted widespread attention and several improved versions have been proposed. For example, Yu et al. introduced different mutation strategies into the growth period and maturity period to increase population diversity [20] and proposed multiple different generation strategies to increase the global search ability [21]. Additionally, they also analyzed the effects of various operations and parameter settings on the performance of the conventional VEGE [22]. Although many pieces in the literature have shown the effectiveness of VEGE, there is still much room for improvement.

The main objective of this paper is to introduce two new search strategies to further improve the search efficiency of VEGE and propose an improved VEGE to better balance search efficiency and population diversity. More specifically, the first strategy, i.e., the dynamic maturity strategy, appropriately introduces competition among individuals to generate more potential seed individuals. In this strategy, we consider the concept of the proximate optimal principle (POP), which suggests that well-performing solutions have similar structures. Therefore, we dynamically allocate computational resources to plants in the seeding operator, where the better individuals can generate more seeds and vice versa. In the second strategy, i.e., the diverse mutation strategy, we observe the relatively weak capacity of VEGE for getting rid of local optima, and the ensemble of the mutation module can improve the ability to escape from trapped local areas. In the numerical experiments, the 30-D and 50-D CEC2013 benchmark functions are employed to evaluate the overall optimization performance of our proposed improved VEGE, and seven engineering problems are adopted to investigate the capacity of our proposal in real-world scenarios. Seven advanced EAs and the conventional VEGE are the competitor algorithms. In addition, we also investigate contributions of the two strategies to performance improvement and their application scenarios. Finally, we provide some open topics for free discussion.

The remainder of this paper is organized as follows. We briefly introduce the optimization framework of the conventional VEGE in Section 2 and give a detailed description of two proposed strategies in Section 3. The parameter settings of the analysis experiment are given in Section 4. We then analyze the effectiveness of the two proposed strategies as well as their strengths and weaknesses in Section 5. Finally, we conclude our work in Section 6.

## 2. Vegetation Evolution

Since the genetic algorithm (GA) [23,24] has sparked a wave of research in the EC community, many well-known EC algorithms have been proposed one after another. Initially, practitioners borrowed biological evolution or the intelligent behavior of animal groups to design new EC algorithms. Then, many natural phenomena as well as human culture have also become sources of inspiration and developed various novel EC algorithms. However, only a few of these derive inspiration from plants to propose new algorithms, such as the flower pollination algorithm [25] and dandelion optimizer [26]. Fortunately, there has also recently been attention to developing new algorithms inspired by the behavior of different plants. As one of the latest members of this branch, the conventional VEGE simulates the common life cycle of plants derived from observations of different plants rather than simulating the behavior of a particular plant to find the global optimum.

Similarly to most heuristic evolutionary algorithms, the conventional VEGE is also population-based and iteratively improves the accuracy of individuals (candidate solutions) to converge to the global optimum. Here, a real plant is modeled as an individual, and each individual sequentially goes through two distinct life periods: growth and maturity. Among these, the growing individuals are responsible for local search, while the mature individuals are responsible for global search. Thus, the unique contribution of the conventional VEGE is to interactively switch between two different search capabilities to balance exploitation and exploration well. A visual demonstration of the the conventional VEGE is shown in Figure 1.

The general optimization process of the conventional VEGE can be summarized simply as follows. Usually, random initialization is used to generate an initial population consisting of multiple individuals. Equation (Equation 1) represents the process of random initialization.
(1)$$\begin{array}{c} P=\left[\begin{array}{c} X_{1}\\ X_{2}\\ X_{3}\\ \vdots \\ X_{n} \end{array}\right]=\left[\begin{array}{cccc} x_{11} & x_{12} & \cdots & x_{1m}\\ x_{21} & x_{22} & \cdots & x_{2m}\\ x_{31} & x_{32} & \cdots & x_{3m}\\ \vdots & \vdots & \ddots & \vdots \\ x_{n1} & x_{n2} & \cdots & x_{nm} \end{array}\right]\\ x_{ij}=r_{1}\times \left(UB_{j}-LB_{j}\right)+LB_{j} \end{array}$$
where $LB_{j}$ and $UB_{j}$ are the lower and upper bounds of the $j^{\mathrm{th}}$ dimension; *n* and *m* are the population size and the dimensionality of the optimization problem, respectively; and $r_{1}$ is a random number in (0, 1). These randomly initialized individuals first enter the growth period, and each individual generates only one offspring individual via Equation (Equation 2).
(2)$$\begin{array}{c} \begin{array}{c} {\vec{X}}_{i}^{t+1}={\vec{X}}_{i}^{t}+\vec{\mathrm{GR}}⊙\vec{\mathrm{GD}} \end{array} \end{array}$$
where $\vec{\mathrm{GR}}$ is a vector to denote the growth radius of plants in every dimension, and $\vec{\mathrm{GD}}$ in (−1, 1) represents the growth direction of plants in every dimension. If the generated offspring individual is better than its parent individual, it will replace the parent individual, otherwise, the parent individual is directly copied to the next generation. After going through a number of growths, i.e., reaching a predefined maximum number of growths, all individuals enter the maturity period. Then, each individual will generate multiple seed individuals by the DE/cur/1-like mutation strategy [17] in Equation (Equation 3). Note that this operation will only be performed once, and the individuals that survive to the next generation will enter the growth period again.
(3)$$\begin{array}{c} {\vec{X}}_{seed}={\vec{X}}_{i}+\vec{\mathrm{MS}}⊙\left({\vec{X}}_{r1}-{\vec{X}}_{r2}\right) \end{array}$$
where $\vec{\mathrm{MS}}$ is the moving scale and is set as a random vector in (−2, 2) to simulate the uncertainty from real environments such as the wind, water flow, and animal behaviors. ${\vec{X}}_{r1}$ and ${\vec{X}}_{r2}$ are two mutually different individuals from ${\vec{X}}_{i}$. Subsequently, all current individuals and seed individuals are mixed to select the best *n* (*n*: population size) individuals to enter the next generation according to their fitness ranking. Finally, these selected individuals will start a new cycle, that is, go through two different periods again until a termination condition is satisfied.

## 3. Our Proposal: VEGE with Dynamic Maturity Strategy and Diverse Mutation Strategy

The conventional VEGE extracts and simulates some common characteristics of the plant life cycle to gradually update the population, where each individual is treated equally and allocated the same resources, such as the same number of growths and seeds. Actually, the real plant ecosystem is much more complex than observed, and differences exist between different species and even between different individuals of the same species. Here, we introduce two new search strategies, i.e., the dynamic maturity and diverse mutation strategies, into the conventional VEGE to simulate the survival mode of real plants more realistically. To better understand the procedures of our proposal, the flowchart is visualized in Figure 2.

### 3.1. Dynamic Maturity Strategy

Due to different living environments, plants need cooperation to ensure the continuation of species, and they also often face competition for survival resources, such as space and nutrients. The diversity of individuals enables them to adapt to changing environments and not be eliminated by nature. However, the conventional VEGE divides all resources equally, and each individual generates exactly the same number of seed individuals. Based on the competitive relationship that exists in nature, we introduce the competitive relationship into the conventional VEGE to reasonably allocate the number of seed individuals generated by each individual.

Suppose the total number of seed individuals that can be generated in each generation is SI. We employ the proposed dynamic maturity strategy to assign the number of seed individuals for each individual in the maturity period. First, each individual will generate the same number of *k* seed individuals, and then the remaining (SI−k×n) seed individuals will be allocated in a competitive manner based on the fitness of all individuals. Here, we use Equation (Equation 4) to determine the probability of each individual being selected and use a roulette wheel to allocate the remaining seed resources. Finally, individuals with better fitness have a higher probability of generating more seed individuals.
(4)$$\begin{array}{c} p_{i}=\mathrm{softmax}\left(\frac{1}{f_{i}}\right)=\frac{\exp(\frac{1}{f_{i}})}{\sum_{j=1}^{n}\exp\left(\frac{1}{f_{i}}\right)} \end{array}$$
where $p_{i}$ is the probability that the *i*-th individual is selected, and $f_{i}$ is the fitness of the *i*-th individual. Algorithm 1 describes the pseudocode of this strategy.
**Algorithm 1** Dynamic maturity strategy  1:Obtain the current population *X* and the fitness value *f*.  2:Initialize the seeding resources SR=[k,k,…,k] for each plant.  3:Calculate the probability *p* for each plant by Equation (Equation 4).  4:Calculate the cumulative probability CP.  5:**for** i=0,…,(SI−k×n) **do**  6:    Generate a random value *r* in (0, 1).  7:    **for** j=1,…,*n* **do**  8:       **if** $CP_{j-1}\leq r<CP_{j}$ **then**  9:           $SR_{j-1}\leftarrow SR_{j-1}+1$.10:       **end if**11:    **end for**12:**end for**13:Output the allocated seeding resources SR.

Since we are taking the minimization problem as an example, the probability is calculated by taking the inverse of fitness. For maximization problems, fitness can be used directly.

### 3.2. Diverse Mutation Strategy

While mutation is an important method for increasing population diversity, the conventional VEGE did not introduce any mutation strategy when it was originally designed. Later, some practitioners realized this defect and adopted different mutation methods to simulate external mutation and internal mutation for individuals in growth and maturity periods, respectively [20]. However, there are various ways of mutation in nature since the complexity of the ecosystem is far beyond our imagination. Thus, we introduce multiple different mutation methods only for seed individuals to provide more diversified potential individuals. In other words, each newly generated seed individual will randomly select one of the following three methods with a uniform probability. Note that these mutations are performed on the seed individuals before their evaluation.

•The genes of a seed individual add a Gaussian noise with a 10% probability. Here, the Gaussian noise is generated by multiplying a standard Gaussian noise by 0.05× (search upper bound—search lower bound).•The seed individual mutates with its parent by a crossover operator with equal probability.•The genes of a seed individual are replaced with a random value in the global space with a probability of 1%.

Algorithm 2 describes the procedure of this diverse mutation strategy.
**Algorithm 2** Diverse mutation strategy  1:**for** i=0,…,SI **do**  2:    Generate a random value *r* in (0, 1).  3:    **if** r<1/3 **then**  4:      **for** j=0,…,*D* **do**  5:         Generate a random value θ in (0, 1).  6:         **if** θ<0.1 **then**  7:           Generate a random value ξ, which follows the normal distribution N(0,1).  8:           $X_{seed,j}=X_{seed,j}+0.05\times \xi \times \left(UB_{j}-LB_{j}\right)$. % Mutation strategy 1  9:         **end if**10:      **end for**11:    **else if** 1/3≤r<2/3 **then**12:      **for** j=0,…,*D* **do**13:         Generate a random value θ in (0, 1).14:         **if** θ<0.5 **then**15:           $X_{seed,j}=X_{seed,j}$.16:         **else**17:           $X_{seed,j}=X_{parent,j}$. % Mutation strategy 218:         **end if**19:      **end for**20:    **else**21:      Generate a random value θ in (0, 1).22:      **if** θ<0.01 **then**23:         Generate a random value ς in (0, 1).24:         $X_{seed,j}=LB_{j}+\varsigma \times \left(UB_{j}-LB_{j}\right)$. % Mutation strategy 325:      **end if**26:    **end if**27:**end for**28:Output seeds after diverse mutation strategies.

So far, the two proposed strategies have been explained, and they are both improvements for the maturity period. Algorithm 3 gives the general framework of the conventional VEGE combined with the two proposed strategies.
**Algorithm 3** The general framework of the conventional VEGE combined with the two proposed strategies. Steps 8 and 10 are our proposed new strategies, respectively  1:Initialize the population randomly.  2:Evaluate the fitness of all initial individuals.  3:**if** all individuals are in the growth period **then**  4:    **for** i=1,…,*n* **do**  5:       Perform the growth operation for *i*-th individual using the method of the conventional VEGE.  6:    **end for**  7:**else**  8:    Determine the number of seed individuals for each individual by Algorithm 1.  9:    All individuals generate seed individuals in turn.10:    Use the proposed diverse mutation strategy in Algorithm 2 for generated seed individuals.11:    Evaluate all seed individuals after undergoing mutation.12:    Mix the current population and all seed individuals to select the next generation.13:**end if**14:Output the found global optimum.

## 4. Experimental Evaluations

Three subsections are involved: benchmark functions, competitor algorithms and the parameter setting, and experimental results.

### 4.1. Benchmark Functions

Since the 28 benchmark functions from the CEC2013 test suite [27] have most of the common features, we run (the conventional VEGE + two strategies), the conventional VEGE, DE, PSO, DE with self-adaptive populations (DE-SAP), phasor PSO (PPSO), social ski-driver optimization (SSDO), RIME, and snow ablation optimization (SAO) on two different dimensions of these functions to analyze the effectiveness of our proposed two strategies. Table 1 gives a detailed summary of the CEC2013 benchmark functions including the multimodal, asymmetry, non-separable variables, and the global optimum value.

Moreover, we investigate the robustness of our proposal in real-world applications. Seven popular engineering optimization problems are employed as test functions, which are listed in Table 2 and were provided by the ENOPPY library [28].

Given that the original versions of all techniques cannot solve the constrained optimization problems, we equip all EAs with the static penalty function [29], which is defined by Equation (Equation 5)
(5)$$\begin{array}{c} F\left(R_{i}\right)=f\left(R_{i}\right)+w\times \sum_{i=1}^{m}\left(\max\left(0,g_{i}\left(R_{i}\right)\right)\right) \end{array}$$
where F(·) is the fitness function, while f(·) and $g_{i}\left(\cdot \right)$ are the objective function and constraint function, respectively. *w* is a constant set to $10^{7}$ by default in the ENOPPY library.

### 4.2. Competitor Algorithms and Parameter Settings

To ensure the fairness of the comparison, we use the number of fitness evaluations to terminate the comparative experiments, and the maximum number is set to 1000×
*dimension* for the CEC2013 28 benchmark functions and 20,000 for engineering problems. In addition, each dimension of each function is run 30 times independently to avoid randomness. Table 3 shows the parameter settings of eight EC algorithms, and the parameter configuration of the two VEGE algorithms is kept exactly the same. In addition, all parameters of the competitor algorithms follow the recommended setting from the corresponding paper, respectively, and the compared DE, PSO, DE-SAP, PPSO, and SSDO are provided by the MEALPY library [30].

### 4.3. Experimental Results

Table 4 and Table 5 give the experimental and statistical results of the numerical experiments on the 30-D and 50-D CEC2013 benchmark functions. We applied the Kruskal–Wallis test and Holm’s multiple comparison test to check whether there is a significant difference between these at the termination of the competitor algorithms. +, ≈, and − are applied to represent that our proposal is significantly better, with no significance, and significantly worse with the compared method, and the best value is in bold. Due to the limitation of space, the optimization convergence curves of the representative functions are provided in Figure 3 and Figure 4. Table 6 provides the detailed optimization results on seven engineering problems, and the convergence curves of the engineering problems are presented in Figure 5. Table 7 and Table 8 summarize the ablation experimental results to investigate the respective contributions of our proposed two strategies to performance improvement. For the sake of simplicity, VEGE + dynamic maturity strategy is referred to as VEGE-i, VEGE + diverse mutation strategy is referred to as VEGE-ii, and VEGE + two strategies is referred to as the Proposal.

## 5. Discussion

We first want to discuss the new benefits of the two strategies. The first strategy, i.e., the dynamic maturity strategy, takes both fixed allocation and dynamic allocation into account so as to allocate the limited number of seed individuals more reasonably. When *k* is set to 0, the number of seed individuals that can be generated by each individual is dynamically allocated according to the fitness of individuals. When *k* is set to $\frac{SI}{n}$, the seed individuals are assigned in the same way as the conventional VEGE. Thus, the allocation method of the conventional VEGE can be seen as a special case of the proposed strategy. We can also adjust the value of *k* to assign different proportions to the two allocation methods, which means that the strategy can flexibly handle various optimization problems with different characteristics. Moreover, the strategy can give the better individuals more opportunities to search space while ensuring that the poorer individuals will not lose the opportunity to continue searching.

The second strategy, i.e., the diverse mutation strategy, provides a variety of different mutations to increase the diversity of the population, and each mutation method modifies the genes of seed individuals with different probabilities. Moreover, the seed individuals generated from the same individual have the opportunity to perform different mutation methods, which can explore more diversified local areas. Especially when the population stagnates, it is helpful to escape from trapped local areas. Since these mutation operations are performed before the fitness evaluation, the second strategy does not add additional fitness consumption. Meanwhile, the CPU consumption resulting from both proposed strategies is also negligible, but the performance improvement is indeed significant. Thus, they can be attributed to low cost and high return.

Next, we want to discuss the potential of the proposed two strategies and provide some open topics. Not limited to the conventional VEGE, our proposal can be easily combined with other improved versions of the VEGE. Since the two strategies are separable, we can also use either of them to combine with VEGE. Thus, a topic worthy of further research is to dynamically select the combination of strategies according to the characteristics of the optimization problem. We simply used three different mutation methods to simulate the mutation patterns of real plants, which is far from enough because the real situation is more diverse and complex. Thus, how to add more diverse mutation methods is also one of our future topics. Since the distribution of individuals is constantly changing with the convergence of the population, the probabilities of different mutation methods being executed should also be different. Therefore, how to efficiently use mutations to guide the convergence of the algorithm is also a promising topic. Although we have only given a few topics, there are still many other interesting topics and hope to give some inspiration to the latecomers.

Subsequently, we apply the Kruskal–Wallis test and Holm’s multiple comparison test to check for significant differences among the compared algorithms on the CEC2013 benchmark functions. The results of the statistical tests show that our two proposed strategies can further improve the performance of the conventional VEGE on most optimization problems, and the deterioration situation is rare and only happens in 50-D $f_{2}$ and $f_{4}$, which fully proves the effectiveness of our two proposed strategies. Moreover, from the experimental and statistical results compared with optimization algorithms in Table 4 and Table 5, our proposal is quite competitive with these state-of-the-art optimization algorithms, and this conclusion can be also observed from the convergence curve of the optimization processes. Due to the population size in VEGE, our proposal being variable, and the initial population size of our proposal being 10 while the other compared algorithms are set to 100, the beginning of the convergence curve in VEGE and our proposal is worse than that of other algorithms, but the excellent exploration and exploitation ability drives our proposal to outperform the compared algorithms rapidly, and the final optimum found by our proposal is also superior, which shows the domination of our proposal over the competitor algorithms in practice.

However, in some multimodal functions, our proposed VEGE with two strategies is significantly inferior to RIME (e.g., 30-D $f_{7}$ and $f_{9}$), and we attempt to explain this degeneration by the No Free Lunch Theory (NFL) [36]. The NFL states that any pair of black-box optimization algorithms has an identical averaging performance in all possible problems, and, if an algorithm performs well on a certain category of problems, it must degenerate on the rest of the problems since it is the only way to achieve identical averaging performance. Although the improvement from VEGE to our proposal can be observed, the original skeleton limits the performance of VEGE in these functions, and we will further analyze the reasons for the failure and give corresponding countermeasures in our future work.

In addition, the ablation experiment results in Table 7 and Table 8 show that the second strategy brings a greater performance improvement than the first strategy on many functions, and the difference between the combination of the two and the second strategy alone is not obvious. This also supports our previous topic, that is, how to reasonably select search strategies is one of the important means by which to improve performance.

Finally, we apply our proposal to simulate the optimization of engineering problems. In real-world applications, another performance indicator that we are concerned about is the robustness of the algorithm. Because evaluation of a real-world problem may be computationally expensive, we hope that the optima found by each trial run will be acceptable and close. Under the identical limitation of fitness evaluations, our proposal can outperform the compared methods significantly, and the mean, the std, the best, and the worst support the robustness of our proposal adequately, which practically proves that our proposal has great potential to deal with real-world optimization problems.

## 6. Conclusions

We introduce two new strategies into the conventional VEGE to further improve performance. The first strategy uses the competitive relationship to rationally allocate resources and expects to generate potential individuals, while the second strategy provides a variety of mutation methods to increase diversity and the ability to escape from local areas. The experimental results confirmed that both of the proposed strategies are effective, and the performance gains become more pronounced as the dimensionality increases.

We will continue to analyze the ecosystem of real plants and use new findings to continuously improve the performance of the conventional VEGE. In future research, we will focus on extending VEGE to various complex optimization tasks such as multi-objective problems [37,38], multimodal problems [39,40], large-scale global optimization problems [41,42], expensive optimization problems [43,44], and feature selection tasks [45,46]. Furthermore, we will also try to apply these improved algorithms to various real-world optimization problems [47,48].

## Figures and Tables

**Figure 1 biomimetics-08-00454-f001:**
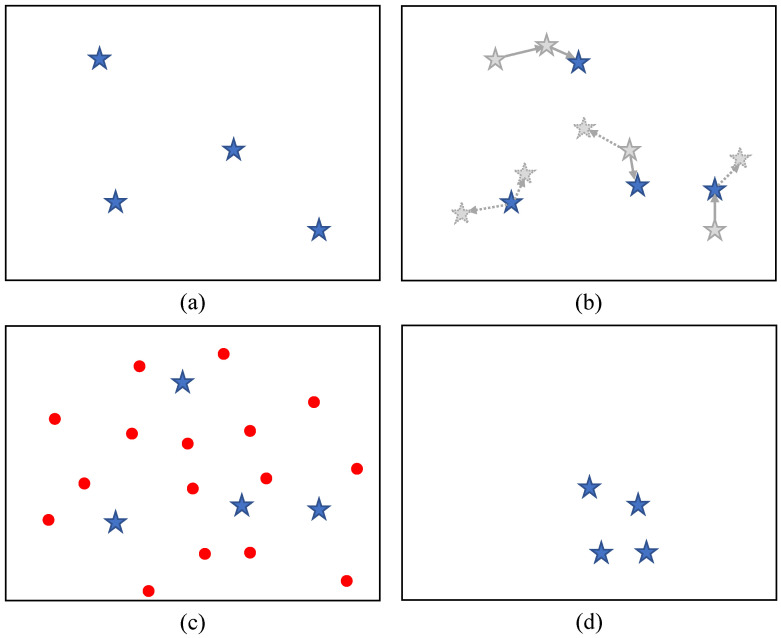
The optimization process of the conventional VEGE, and (**a**–**d**) demonstrate initialization, growth period, maturity period, and selection, respectively. The dashed arrows indicate the growth vector of individuals, and all red dots indicate generated seed individuals.

**Figure 2 biomimetics-08-00454-f002:**
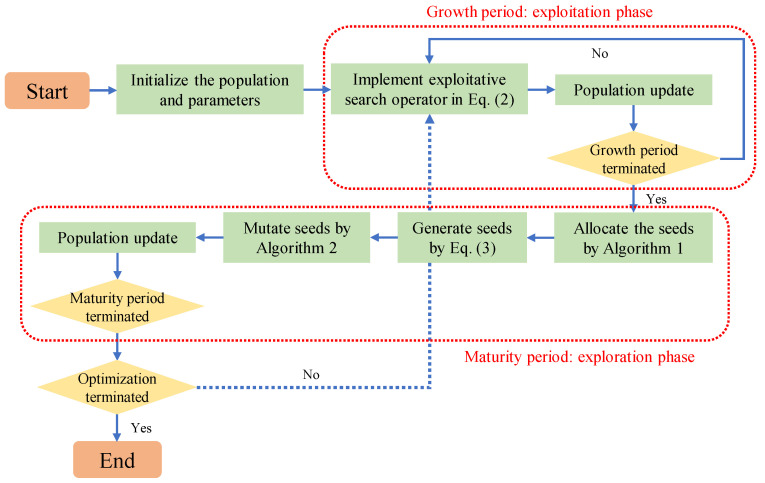
The flowchart of our proposal.

**Figure 3 biomimetics-08-00454-f003:**
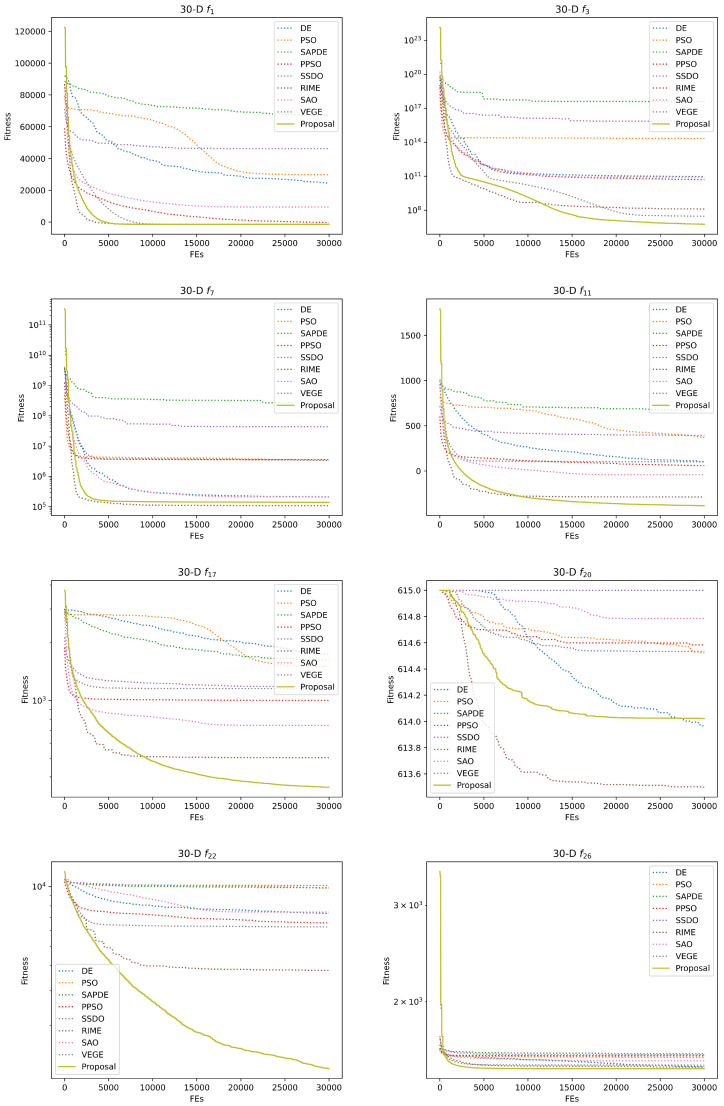
Convergence curves of competitor algorithms on 30-D CEC2013 representative benchmark functions ($f_{1}$ and $f_{3}$: unimodal functions; $f_{7}$, $f_{11}$, $f_{17}$, and $f_{20}$: multimodal functions; $f_{22}$ and $f_{26}$: composite functions).

**Figure 4 biomimetics-08-00454-f004:**
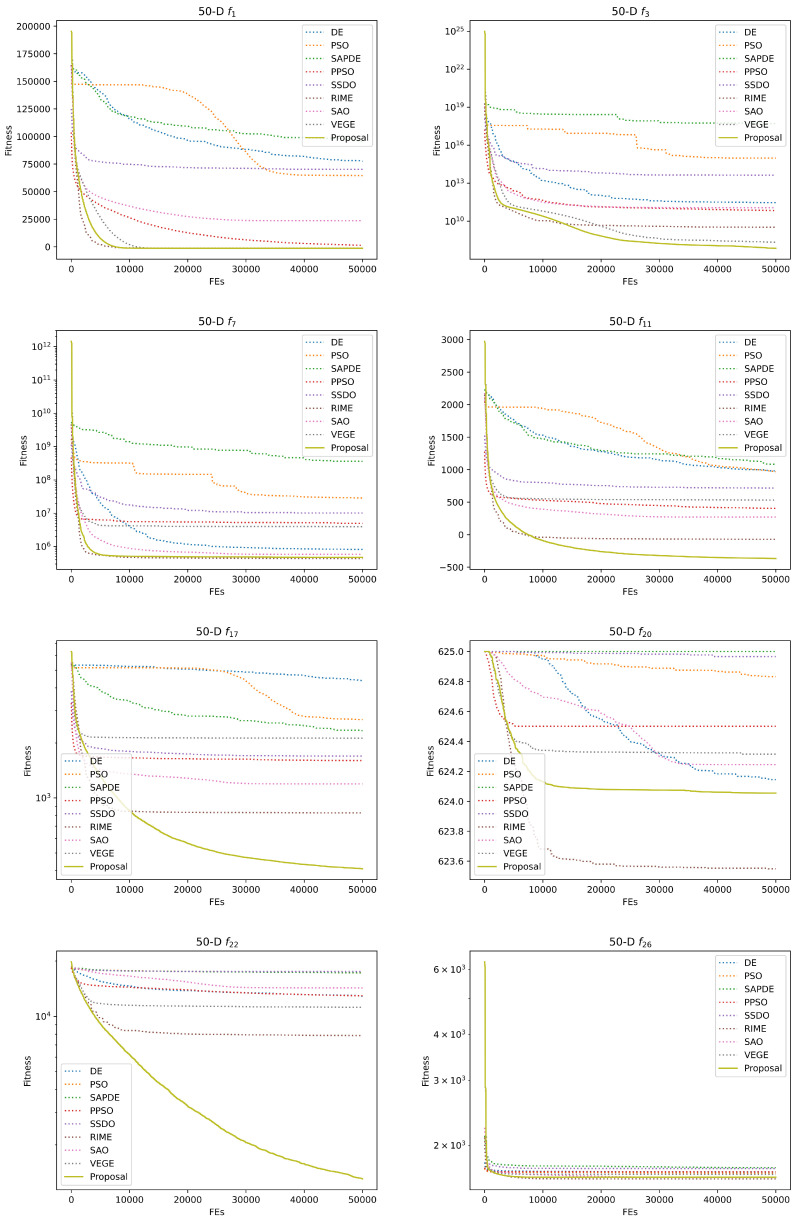
Convergence curves of competitor algorithms on 50-D CEC2013 representative benchmark functions.

**Figure 5 biomimetics-08-00454-f005:**
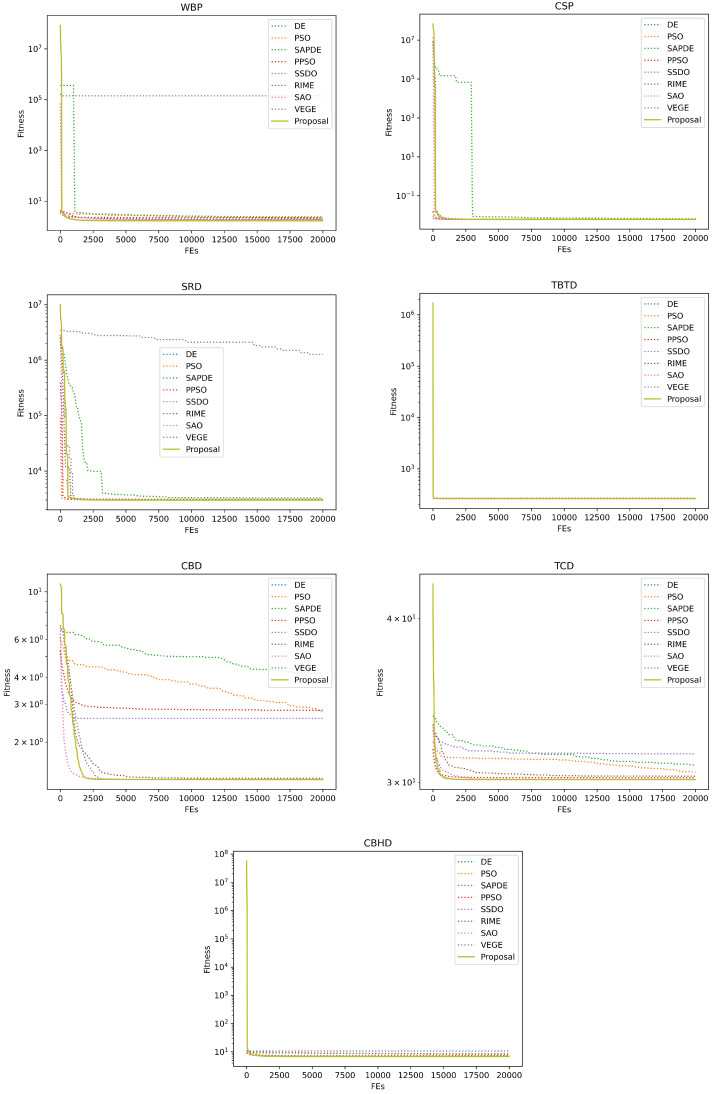
Convergence curves of competitor algorithms on seven engineering optimization problems.

**Table 1 biomimetics-08-00454-t001:** Summary of the CEC2013 suite: Uni. = unimodal function, Multi. = multimodal function, Comp. = composition function. Search space for each function is [−100, 100]${}^{D}$.

Fun.	Description	Feature	Optimum
$f_{1}$	Sphere Function	Uni.	−1400
$f_{2}$	Rotated High Conditioned Elliptic Function		−1300
$f_{3}$	Rotated Bent Cigar Function		−1200
$f_{4}$	Rotated Discus Function		−1100
$f_{5}$	Different Powers Function		−1000
$f_{6}$	Rotated Rosenbrock’s Function	Multi.	−900
$f_{7}$	Rotated Schaffers F7 Function		−800
$f_{8}$	Rotated Ackley’s Function		−700
$f_{9}$	Rotated Weierstrass Function		−600
$f_{10}$	Rotated Griewank’s Function		−500
$f_{11}$	Rastrigin’s Function		−400
$f_{12}$	Rotated Rastrigin’s Function		−300
$f_{13}$	Non-Continuous Rotated Rastrigin’s Function		−200
$f_{14}$	Schwefel’s Function		−100
$f_{15}$	Rotated Schwefel’s Function		100
$f_{16}$	Rotated Katsuura Function		200
$f_{17}$	Lunacek Bi-Rastrigin Function		300
$f_{18}$	Rotated Lunacek Bi-Rastrigin Function		400
$f_{19}$	Expanded Griewank’s plus Rosenbrock’s Function		500
$f_{20}$	Expanded Scaffer’s F6 Function		600
$f_{21}$	Composition Function 1 (n = 5, Rotated)	Comp.	700
$f_{22}$	Composition Function 2 (n = 3, Unrotated)		800
$f_{23}$	Composition Function 3 (n = 3, Rotated)		900
$f_{24}$	Composition Function 4 (n = 3, Rotated)		1000
$f_{25}$	Composition Function 5 (n = 3, Rotated)		1100
$f_{26}$	Composition Function 6 (n = 5, Rotated)		1200
$f_{27}$	Composition Function 7 (n = 5, Rotated)		1300
$f_{28}$	Composition Function 8 (n = 5, Rotated)		1400
Search space: [−100, 100]${}^{D}$			

**Table 2 biomimetics-08-00454-t002:** Summary of seven engineering optimization problems: Dim. = dimension size.

Name	Abbr.	Dim.	The Number of Constraints
Welded Beam Problem	WBP	4	7
Compression Spring Problem	CSP	4	4
Speed Reducer Problem	SRD	7	11
Three Bar Truss Problem	TBTD	2	3
Cantilever Beam Problem	CBD	5	1
Tubular Column Problem	TCD	2	6
Corrugated Bulkhead Problem	CBHD	4	6

**Table 3 biomimetics-08-00454-t003:** The parameter settings of eight comparison algorithms.

EAs	Parameters	Value
DE (1995) [17]	Population size	100
	Scaling factor *F*	1
	Crossover rate Cr	0.9
	Mutation strategy	DE/cur-to-rand/1/bin
PSO (1995) [18]	Population size	100
	Inertia factor *w*	1
	Coefficients $c_{1}$ and $c_{2}$	2.05
	Max. and min. speed	2, −2
DE-SAP (2006) [31]	Population size	100
	Encoding method *w*	absolute encoding (Abs)
PPSO (2019) [32]	Population size	100
SSDO (2020) [33]	Population size	100
VEGE (2022) [16]	Population size	10
	Growth cycle GC	6
	Growth radius GR	2
	Growth direction GD	a random number in [−1,1]
	Total # of seed individuals	60
	Moving scaling MS	a random number in [−2,2]
RIME (2023) [34]	Population size	100
	parameter *w*	5
SAO (2023) [35]	Population size	100

**Table 4 biomimetics-08-00454-t004:** Experimental and statistical results on 30-D CEC2013 benchmark functions. $f_{1}–f_{5}$: unimodal functions; $f_{6}–f_{20}$: basic multimodal functions; $f_{21}–f_{28}$: composition functions (Proposal: VEGE + two proposed strategies).

Func.		DE	PSO	DE-SAP	PPSO	SSDO	RIME	SAO	VEGE	Proposal
$f_{1}$	mean	2.44$\;\times \;10^{4}$ +	2.98$\;\times \;10^{4}$ +	6.71$\;\times \;10^{4}$ +	−2.71$\;\times \;10^{2}$ +	4.62$\;\times \;10^{4}$ +	−1.38$\;\times \;10^{3}$ +	9.49$\;\times \;10^{3}$ +	−1.40$\;\times \;10^{3}$ +	**−1.40$\;\times \;10^{3}$**
	std	2.97$\;\times \;10^{3}$	1.14$\;\times \;10^{4}$	8.84$\;\times \;10^{3}$	6.02$\;\times \;10^{2}$	3.06$\;\times \;10^{3}$	6.43$\;\times \;10^{0}$	4.06$\;\times \;10^{3}$	8.37$\;\times \;10^{-1}$	**1.63$\;\times \;10^{-1}$**
$f_{2}$	mean	2.79$\;\times \;10^{8}$ +	3.82$\;\times \;10^{8}$ +	1.25$\;\times \;10^{9}$ +	5.84$\;\times \;10^{7}$ +	6.36$\;\times \;10^{8}$ +	2.62$\;\times \;10^{7}$ +	1.04$\;\times \;10^{8}$ +	5.43$\;\times \;10^{6}$ ≈	**5.35$\;\times \;10^{6}$**
	std	8.22$\;\times \;10^{7}$	2.54$\;\times \;10^{8}$	4.14$\;\times \;10^{8}$	2.56$\;\times \;10^{7}$	1.25$\;\times \;10^{8}$	1.15$\;\times \;10^{7}$	4.91$\;\times \;10^{7}$	2.36$\;\times \;10^{6}$	**2.50$\;\times \;10^{6}$**
$f_{3}$	mean	9.14$\;\times \;10^{10}$ +	2.19$\;\times \;10^{14}$ +	3.97$\;\times \;10^{17}$ +	4.97$\;\times \;10^{10}$ +	7.34$\;\times \;10^{15}$ +	1.27$\;\times \;10^{8}$ +	8.36$\;\times \;10^{10}$ +	2.98$\;\times \;10^{7}$ +	**5.73$\;\times \;10^{6}$**
	std	1.12$\;\times \;10^{10}$	8.96$\;\times \;10^{14}$	1.64$\;\times \;10^{18}$	2.91$\;\times \;10^{10}$	1.11$\;\times \;10^{16}$	1.52$\;\times \;10^{8}$	5.66$\;\times \;10^{10}$	4.89$\;\times \;10^{7}$	**9.05$\;\times \;10^{6}$**
$f_{4}$	mean	1.57$\;\times \;10^{5}$ +	1.95$\;\times \;10^{5}$ +	1.18$\;\times \;10^{5}$ +	7.54$\;\times \;10^{4}$ +	6.27$\;\times \;10^{4}$ +	3.29$\;\times \;10^{4}$ +	5.73$\;\times \;10^{4}$ +	1.54$\;\times \;10^{4}$ ≈	**1.39$\;\times \;10^{4}$**
	std	1.85$\;\times \;10^{4}$	6.80$\;\times \;10^{4}$	2.32$\;\times \;10^{4}$	1.61$\;\times \;10^{4}$	2.21$\;\times \;10^{3}$	1.03$\;\times \;10^{4}$	6.35$\;\times \;10^{3}$	6.52$\;\times \;10^{3}$	**5.15$\;\times \;10^{3}$**
$f_{5}$	mean	6.44$\;\times \;10^{3}$ +	5.19$\;\times \;10^{4}$ +	7.78$\;\times \;10^{4}$ +	6.52$\;\times \;10^{2}$ +	4.23$\;\times \;10^{4}$ +	−8.18$\;\times \;10^{2}$ +	7.37$\;\times \;10^{3}$ +	−9.97$\;\times \;10^{2}$ +	**−9.99$\;\times \;10^{2}$**
	std	1.23$\;\times \;10^{3}$	1.79$\;\times \;10^{4}$	2.44$\;\times \;10^{4}$	1.91$\;\times \;10^{3}$	8.02$\;\times \;10^{3}$	5.99$\;\times \;10^{1}$	4.18$\;\times \;10^{3}$	4.32$\;\times \;10^{0}$	**2.07$\;\times \;10^{-1}$**
$f_{6}$	mean	1.12$\;\times \;10^{3}$ +	4.26$\;\times \;10^{3}$ +	1.35$\;\times \;10^{4}$ +	−6.05$\;\times \;10^{2}$ +	7.16$\;\times \;10^{3}$ +	−8.19$\;\times \;10^{2}$ +	−2.77$\;\times \;10^{2}$ +	−8.32$\;\times \;10^{2}$ ≈	**−8.40$\;\times \;10^{2}$**
	std	4.32$\;\times \;10^{2}$	3.41$\;\times \;10^{3}$	4.60$\;\times \;10^{3}$	1.20$\;\times \;10^{2}$	1.04$\;\times \;10^{3}$	2.88$\;\times \;10^{1}$	2.39$\;\times \;10^{2}$	3.04$\;\times \;10^{1}$	**3.62$\;\times \;10^{1}$**
$f_{7}$	mean	2.10$\;\times \;10^{5}$ +	3.38$\;\times \;10^{6}$ +	1.83$\;\times \;10^{8}$ +	3.58$\;\times \;10^{6}$ +	4.37$\;\times \;10^{7}$ +	**1.08$\;\times \;10^{5}$** −	2.14$\;\times \;10^{5}$ +	3.42$\;\times \;10^{6}$ +	1.40$\;\times \;10^{5}$
	std	1.32$\;\times \;10^{4}$	4.68$\;\times \;10^{6}$	1.85$\;\times \;10^{8}$	5.59$\;\times \;10^{6}$	2.89$\;\times \;10^{7}$	**1.66$\;\times \;10^{4}$**	1.74$\;\times \;10^{5}$	1.24$\;\times \;10^{7}$	4.08$\;\times \;10^{4}$
$f_{8}$	mean	**−6.79$\;\times \;10^{2}$** ≈	−6.79$\;\times \;10^{2}$ ≈	−6.79$\;\times \;10^{2}$ ≈	−6.79$\;\times \;10^{2}$ ≈	−6.79$\;\times \;10^{2}$ ≈	−6.79$\;\times \;10^{2}$ ≈	−6.79$\;\times \;10^{2}$ ≈	−6.79$\;\times \;10^{2}$ ≈	−6.79$\;\times \;10^{2}$
	std	**6.05$\;\times \;10^{-2}$**	4.48$\;\times \;10^{-2}$	5.20$\;\times \;10^{-2}$	7.12$\;\times \;10^{-2}$	6.03$\;\times \;10^{-2}$	6.93$\;\times \;10^{-2}$	8.65$\;\times \;10^{-2}$	4.44$\;\times \;10^{-2}$	4.68$\;\times \;10^{-2}$
$f_{9}$	mean	−5.59$\;\times \;10^{2}$ +	−5.62$\;\times \;10^{2}$ +	−5.57$\;\times \;10^{2}$ +	−5.61$\;\times \;10^{2}$ +	−5.58$\;\times \;10^{2}$ +	**−5.75$\;\times \;10^{2}$** −	−5.70$\;\times \;10^{2}$ ≈	−5.68$\;\times \;10^{2}$ ≈	−5.71$\;\times \;10^{2}$
	std	1.44$\;\times \;10^{0}$	3.92$\;\times \;10^{0}$	1.30$\;\times \;10^{0}$	2.58$\;\times \;10^{0}$	1.01$\;\times \;10^{0}$	**5.03$\;\times \;10^{0}$**	3.17$\;\times \;10^{0}$	3.98$\;\times \;10^{0}$	4.13$\;\times \;10^{0}$
$f_{10}$	mean	2.74$\;\times \;10^{3}$ +	4.82$\;\times \;10^{3}$ +	9.35$\;\times \;10^{3}$ +	−1.02$\;\times \;10^{1}$ +	6.07$\;\times \;10^{3}$ +	−4.46$\;\times \;10^{2}$ +	1.07$\;\times \;10^{3}$ +	−4.96$\;\times \;10^{2}$ +	**−4.97$\;\times \;10^{2}$**
	std	5.09$\;\times \;10^{2}$	2.05$\;\times \;10^{3}$	1.89$\;\times \;10^{3}$	2.76$\;\times \;10^{2}$	6.06$\;\times \;10^{2}$	2.19$\;\times \;10^{1}$	4.90$\;\times \;10^{2}$	6.48$\;\times \;10^{-1}$	**6.01$\;\times \;10^{-1}$**
$f_{11}$	mean	1.04$\;\times \;10^{2}$ +	3.71$\;\times \;10^{2}$ +	6.69$\;\times \;10^{2}$ +	5.85$\;\times \;10^{1}$ +	3.94$\;\times \;10^{2}$ +	−2.88$\;\times \;10^{2}$ +	−4.05$\;\times \;10^{1}$ +	1.03$\;\times \;10^{2}$ +	**−3.84$\;\times \;10^{2}$**
	std	3.34$\;\times \;10^{1}$	1.66$\;\times \;10^{2}$	1.60$\;\times \;10^{2}$	9.32$\;\times \;10^{1}$	4.50$\;\times \;10^{1}$	1.77$\;\times \;10^{1}$	7.24$\;\times \;10^{1}$	1.03$\;\times \;10^{2}$	**4.27$\;\times \;10^{0}$**
$f_{12}$	mean	2.65$\;\times \;10^{2}$ +	4.39$\;\times \;10^{2}$ +	7.95$\;\times \;10^{2}$ +	1.89$\;\times \;10^{2}$ +	4.47$\;\times \;10^{2}$ +	**−1.55$\;\times \;10^{2}$** −	6.56$\;\times \;10^{1}$ ≈	1.54$\;\times \;10^{2}$ +	5.69$\;\times \;10^{1}$
	std	4.69$\;\times \;10^{1}$	1.67$\;\times \;10^{2}$	1.27$\;\times \;10^{2}$	1.09$\;\times \;10^{2}$	3.80$\;\times \;10^{1}$	**3.68$\;\times \;10^{1}$**	7.28$\;\times \;10^{1}$	1.10$\;\times \;10^{2}$	9.84$\;\times \;10^{1}$
$f_{13}$	mean	3.21$\;\times \;10^{2}$ +	5.17$\;\times \;10^{2}$ +	7.94$\;\times \;10^{2}$ +	3.62$\;\times \;10^{2}$ +	5.03$\;\times \;10^{2}$ +	**1.06$\;\times \;10^{1}$** −	1.66$\;\times \;10^{2}$ ≈	3.70$\;\times \;10^{2}$ +	1.91$\;\times \;10^{2}$
	std	3.82$\;\times \;10^{1}$	1.60$\;\times \;10^{2}$	1.62$\;\times \;10^{2}$	9.39$\;\times \;10^{1}$	4.31$\;\times \;10^{1}$	**2.59$\;\times \;10^{1}$**	6.33$\;\times \;10^{1}$	1.24$\;\times \;10^{2}$	7.60$\;\times \;10^{1}$
$f_{14}$	mean	6.03$\;\times \;10^{3}$ +	8.31$\;\times \;10^{3}$ +	8.35$\;\times \;10^{3}$ +	5.18$\;\times \;10^{3}$ +	8.37$\;\times \;10^{3}$ +	2.42$\;\times \;10^{3}$ +	5.32$\;\times \;10^{3}$ +	4.59$\;\times \;10^{3}$ +	**2.49$\;\times \;10^{2}$**
	std	3.85$\;\times \;10^{2}$	2.80$\;\times \;10^{2}$	2.86$\;\times \;10^{2}$	8.16$\;\times \;10^{2}$	2.19$\;\times \;10^{2}$	4.21$\;\times \;10^{2}$	7.98$\;\times \;10^{2}$	4.93$\;\times \;10^{2}$	**1.66$\;\times \;10^{2}$**
$f_{15}$	mean	8.11$\;\times \;10^{3}$ +	8.57$\;\times \;10^{3}$ +	8.46$\;\times \;10^{3}$ +	6.32$\;\times \;10^{3}$ +	7.82$\;\times \;10^{3}$ +	5.08$\;\times \;10^{3}$ +	5.80$\;\times \;10^{3}$ +	4.52$\;\times \;10^{3}$ ≈	**4.20$\;\times \;10^{3}$**
	std	3.64$\;\times \;10^{2}$	4.61$\;\times \;10^{2}$	3.11$\;\times \;10^{2}$	8.14$\;\times \;10^{2}$	2.94$\;\times \;10^{2}$	6.66$\;\times \;10^{2}$	5.23$\;\times \;10^{2}$	4.96$\;\times \;10^{2}$	**5.88$\;\times \;10^{2}$**
$f_{16}$	mean	2.03$\;\times \;10^{2}$ +	2.03$\;\times \;10^{2}$ +	2.03$\;\times \;10^{2}$ +	2.02$\;\times \;10^{2}$ +	2.03$\;\times \;10^{2}$ +	2.02$\;\times \;10^{2}$ +	**2.01$\;\times \;10^{2}$** ≈	2.02$\;\times \;10^{2}$ +	2.01$\;\times \;10^{2}$
	std	3.23$\;\times \;10^{-1}$	3.48$\;\times \;10^{-1}$	3.15$\;\times \;10^{-1}$	5.98$\;\times \;10^{-1}$	3.54$\;\times \;10^{-1}$	5.34$\;\times \;10^{-1}$	**3.59$\;\times \;10^{-1}$**	4.48$\;\times \;10^{-1}$	3.57$\;\times \;10^{-1}$
$f_{17}$	mean	1.75$\;\times \;10^{3}$ +	1.50$\;\times \;10^{3}$ +	1.63$\;\times \;10^{3}$ +	1.00$\;\times \;10^{3}$ +	1.18$\;\times \;10^{3}$ +	5.03$\;\times \;10^{2}$ +	7.43$\;\times \;10^{2}$ +	1.15$\;\times \;10^{3}$ +	**3.54$\;\times \;10^{2}$**
	std	1.68$\;\times \;10^{2}$	2.41$\;\times \;10^{2}$	2.53$\;\times \;10^{2}$	1.01$\;\times \;10^{2}$	3.80$\;\times \;10^{1}$	2.76$\;\times \;10^{1}$	8.99$\;\times \;10^{1}$	1.94$\;\times \;10^{2}$	**5.91$\;\times \;10^{0}$**
$f_{18}$	mean	1.71$\;\times \;10^{3}$ +	2.04$\;\times \;10^{3}$ +	3.31$\;\times \;10^{3}$ +	1.70$\;\times \;10^{3}$ +	2.60$\;\times \;10^{3}$ +	**6.28$\;\times \;10^{2}$** −	1.23$\;\times \;10^{3}$ +	1.92$\;\times \;10^{3}$ +	7.08$\;\times \;10^{2}$
	std	1.23$\;\times \;10^{2}$	4.13$\;\times \;10^{2}$	3.50$\;\times \;10^{2}$	2.82$\;\times \;10^{2}$	1.07$\;\times \;10^{2}$	**3.70$\;\times \;10^{1}$**	1.94$\;\times \;10^{2}$	3.64$\;\times \;10^{2}$	8.71$\;\times \;10^{1}$
$f_{19}$	mean	1.13$\;\times \;10^{5}$ +	2.93$\;\times \;10^{5}$ +	2.95$\;\times \;10^{6}$ +	2.72$\;\times \;10^{3}$ +	6.88$\;\times \;10^{5}$ +	**5.19$\;\times \;10^{2}$** ≈	8.66$\;\times \;10^{3}$ +	5.32$\;\times \;10^{2}$ +	5.19$\;\times \;10^{2}$
	std	5.39$\;\times \;10^{4}$	3.10$\;\times \;10^{5}$	1.44$\;\times \;10^{6}$	5.16$\;\times \;10^{3}$	2.20$\;\times \;10^{5}$	**3.79$\;\times \;10^{0}$**	1.19$\;\times \;10^{4}$	5.45$\;\times \;10^{0}$	6.41$\;\times \;10^{0}$
$f_{20}$	mean	6.14$\;\times \;10^{2}$ ≈	6.15$\;\times \;10^{2}$ +	6.15$\;\times \;10^{2}$ +	6.15$\;\times \;10^{2}$ +	6.15$\;\times \;10^{2}$ +	**6.13$\;\times \;10^{2}$** −	6.15$\;\times \;10^{2}$ +	6.15$\;\times \;10^{2}$ +	6.14$\;\times \;10^{2}$
	std	1.94$\;\times \;10^{-1}$	4.84$\;\times \;10^{-1}$	2.26$\;\times \;10^{-7}$	1.81$\;\times \;10^{-1}$	1.95$\;\times \;10^{-4}$	**7.51$\;\times \;10^{-1}$**	4.62$\;\times \;10^{-1}$	1.96$\;\times \;10^{-1}$	7.55$\;\times \;10^{-1}$
$f_{21}$	mean	4.10$\;\times \;10^{3}$ +	3.71$\;\times \;10^{3}$ +	4.50$\;\times \;10^{3}$ +	2.05$\;\times \;10^{3}$ +	3.20$\;\times \;10^{3}$ +	**1.67$\;\times \;10^{3}$** −	2.59$\;\times \;10^{3}$ +	1.83$\;\times \;10^{3}$ +	1.69$\;\times \;10^{3}$
	std	1.88$\;\times \;10^{2}$	6.52$\;\times \;10^{2}$	5.64$\;\times \;10^{2}$	1.24$\;\times \;10^{2}$	8.10$\;\times \;10^{1}$	**2.63$\;\times \;10^{2}$**	1.85$\;\times \;10^{2}$	2.45$\;\times \;10^{2}$	2.69$\;\times \;10^{2}$
$f_{22}$	mean	7.31$\;\times \;10^{3}$ +	9.77$\;\times \;10^{3}$ +	9.86$\;\times \;10^{3}$ +	6.56$\;\times \;10^{3}$ +	1.01$\;\times \;10^{4}$ +	3.78$\;\times \;10^{3}$ +	7.44$\;\times \;10^{3}$ +	6.26$\;\times \;10^{3}$ +	**1.21$\;\times \;10^{3}$**
	std	3.42$\;\times \;10^{2}$	5.71$\;\times \;10^{2}$	4.61$\;\times \;10^{2}$	8.86$\;\times \;10^{2}$	2.56$\;\times \;10^{2}$	4.96$\;\times \;10^{2}$	9.26$\;\times \;10^{2}$	7.09$\;\times \;10^{2}$	**1.40$\;\times \;10^{2}$**
$f_{23}$	mean	9.06$\;\times \;10^{3}$ +	9.52$\;\times \;10^{3}$ +	9.69$\;\times \;10^{3}$ +	7.59$\;\times \;10^{3}$ +	9.62$\;\times \;10^{3}$ +	6.14$\;\times \;10^{3}$ ≈	7.86$\;\times \;10^{3}$ +	5.86$\;\times \;10^{3}$ ≈	**5.85$\;\times \;10^{3}$**
	std	3.24$\;\times \;10^{2}$	3.09$\;\times \;10^{2}$	3.76$\;\times \;10^{2}$	7.65$\;\times \;10^{2}$	2.87$\;\times \;10^{2}$	9.04$\;\times \;10^{2}$	1.11$\;\times \;10^{3}$	6.87$\;\times \;10^{2}$	**7.33$\;\times \;10^{2}$**
$f_{24}$	mean	1.31$\;\times \;10^{3}$ +	1.31$\;\times \;10^{3}$ +	1.36$\;\times \;10^{3}$ +	1.32$\;\times \;10^{3}$ +	1.36$\;\times \;10^{3}$ +	**1.26$\;\times \;10^{3}$** −	1.29$\;\times \;10^{3}$ +	1.30$\;\times \;10^{3}$ +	1.28$\;\times \;10^{3}$
	std	3.56$\;\times \;10^{0}$	8.40$\;\times \;10^{0}$	1.77$\;\times \;10^{1}$	7.86$\;\times \;10^{0}$	1.35$\;\times \;10^{1}$	**1.02$\;\times \;10^{1}$**	1.28$\;\times \;10^{1}$	1.34$\;\times \;10^{1}$	1.12$\;\times \;10^{1}$
$f_{25}$	mean	1.41$\;\times \;10^{3}$ +	1.42$\;\times \;10^{3}$ +	1.45$\;\times \;10^{3}$ +	1.43$\;\times \;10^{3}$ +	1.47$\;\times \;10^{3}$ +	**1.38$\;\times \;10^{3}$** −	1.42$\;\times \;10^{3}$ +	1.43$\;\times \;10^{3}$ +	1.40$\;\times \;10^{3}$
	std	6.05$\;\times \;10^{0}$	8.92$\;\times \;10^{0}$	9.25$\;\times \;10^{0}$	1.34$\;\times \;10^{1}$	1.52$\;\times \;10^{1}$	**8.43$\;\times \;10^{0}$**	1.31$\;\times \;10^{1}$	1.42$\;\times \;10^{1}$	1.62$\;\times \;10^{1}$
$f_{26}$	mean	1.52$\;\times \;10^{3}$ ≈	1.58$\;\times \;10^{3}$ +	1.61$\;\times \;10^{3}$ +	1.59$\;\times \;10^{3}$ +	1.60$\;\times \;10^{3}$ +	1.52$\;\times \;10^{3}$ ≈	1.56$\;\times \;10^{3}$ +	1.53$\;\times \;10^{3}$ ≈	**1.51$\;\times \;10^{3}$**
	std	7.56$\;\times \;10^{1}$	4.77$\;\times \;10^{1}$	3.51$\;\times \;10^{1}$	5.02$\;\times \;10^{1}$	3.29$\;\times \;10^{1}$	7.11$\;\times \;10^{1}$	5.99$\;\times \;10^{1}$	8.66$\;\times \;10^{1}$	**8.33$\;\times \;10^{1}$**
$f_{27}$	mean	2.71$\;\times \;10^{3}$ +	2.72$\;\times \;10^{3}$ +	3.11$\;\times \;10^{3}$ +	2.75$\;\times \;10^{3}$ +	3.21$\;\times \;10^{3}$ +	**2.26$\;\times \;10^{3}$** −	2.55$\;\times \;10^{3}$ +	2.67$\;\times \;10^{3}$ +	2.43$\;\times \;10^{3}$
	std	2.90$\;\times \;10^{1}$	8.05$\;\times \;10^{1}$	1.33$\;\times \;10^{2}$	7.33$\;\times \;10^{1}$	1.20$\;\times \;10^{2}$	**1.00$\;\times \;10^{2}$**	9.49$\;\times \;10^{1}$	1.12$\;\times \;10^{2}$	8.71$\;\times \;10^{1}$
$f_{28}$	mean	4.91$\;\times \;10^{3}$ +	6.10$\;\times \;10^{3}$ +	7.41$\;\times \;10^{3}$ +	5.04$\;\times \;10^{3}$ +	5.70$\;\times \;10^{3}$ +	**2.32$\;\times \;10^{3}$** −	4.45$\;\times \;10^{3}$ +	4.93$\;\times \;10^{3}$ +	3.64$\;\times \;10^{3}$
	std	2.10$\;\times \;10^{2}$	7.46$\;\times \;10^{2}$	6.90$\;\times \;10^{2}$	5.27$\;\times \;10^{2}$	1.83$\;\times \;10^{2}$	**4.34$\;\times \;10^{2}$**	3.91$\;\times \;10^{2}$	4.48$\;\times \;10^{2}$	1.22$\;\times \;10^{3}$
		+/≈/−: 25/3/0	+/≈/−: 27/1/0	+/≈/−: 27/1/0	+/≈/−: 27/1/0	+/≈/−: 27/1/0	+/≈/−: 13/4/11	+/≈/−: 23/5/0	+/≈/−: 20/8/0	

**Table 5 biomimetics-08-00454-t005:** Experimental and statistical results on 50-D CEC2013 benchmark functions.

Func.		DE	PSO	DE-SAP	PPSO	SSDO	RIME	SAO	VEGE	Proposal
$f_{1}$	mean	7.75$\;\times \;10^{4}$ +	6.46$\;\times \;10^{4}$ +	9.55$\;\times \;10^{4}$ +	1.34$\;\times \;10^{3}$ +	7.03$\;\times \;10^{4}$ +	−1.28$\;\times \;10^{3}$ +	2.36$\;\times \;10^{4}$ +	−1.39$\;\times \;10^{3}$ +	**−1.40$\;\times \;10^{3}$**
	std	7.23$\;\times \;10^{3}$	1.47$\;\times \;10^{4}$	1.11$\;\times \;10^{4}$	9.91$\;\times \;10^{2}$	3.31$\;\times \;10^{3}$	2.69$\;\times \;10^{1}$	5.14$\;\times \;10^{3}$	2.03$\;\times \;10^{0}$	**1.07$\;\times \;10^{0}$**
$f_{2}$	mean	1.11$\;\times \;10^{9}$ +	1.63$\;\times \;10^{9}$ +	3.68$\;\times \;10^{9}$ +	1.47$\;\times \;10^{8}$ +	1.83$\;\times \;10^{9}$ +	7.47$\;\times \;10^{7}$ +	2.06$\;\times \;10^{8}$ +	**9.29$\;\times \;10^{6}$** −	1.15$\;\times \;10^{7}$
	std	1.34$\;\times \;10^{8}$	8.33$\;\times \;10^{8}$	8.69$\;\times \;10^{8}$	4.46$\;\times \;10^{7}$	3.93$\;\times \;10^{8}$	1.95$\;\times \;10^{7}$	6.91$\;\times \;10^{7}$	**2.54$\;\times \;10^{6}$**	2.89$\;\times \;10^{6}$
$f_{3}$	mean	2.92$\;\times \;10^{11}$ +	9.75$\;\times \;10^{14}$ +	5.17$\;\times \;10^{17}$ +	6.95$\;\times \;10^{10}$ +	4.29$\;\times \;10^{13}$ +	3.33$\;\times \;10^{9}$ +	1.19$\;\times \;10^{11}$ +	2.21$\;\times \;10^{8}$ +	**7.26$\;\times \;10^{7}$**
	std	2.16$\;\times \;10^{10}$	3.15$\;\times \;10^{15}$	1.79$\;\times \;10^{18}$	2.88$\;\times \;10^{10}$	4.17$\;\times \;10^{13}$	4.83$\;\times \;10^{9}$	4.18$\;\times \;10^{10}$	4.31$\;\times \;10^{8}$	**1.70$\;\times \;10^{8}$**
$f_{4}$	mean	2.76$\;\times \;10^{5}$ +	2.95$\;\times \;10^{5}$ +	1.85$\;\times \;10^{5}$ +	1.20$\;\times \;10^{5}$ +	8.59$\;\times \;10^{4}$ +	6.49$\;\times \;10^{4}$ +	8.72$\;\times \;10^{4}$ +	**1.06$\;\times \;10^{4}$** −	1.47$\;\times \;10^{4}$
	std	2.24$\;\times \;10^{4}$	9.53$\;\times \;10^{4}$	5.35$\;\times \;10^{4}$	1.95$\;\times \;10^{4}$	3.71$\;\times \;10^{3}$	1.35$\;\times \;10^{4}$	6.38$\;\times \;10^{3}$	**4.20$\;\times \;10^{3}$**	4.77$\;\times \;10^{3}$
$f_{5}$	mean	4.19$\;\times \;10^{4}$ +	1.06$\;\times \;10^{5}$ +	9.11$\;\times \;10^{4}$ +	1.51$\;\times \;10^{3}$ +	3.70$\;\times \;10^{4}$ +	−4.03$\;\times \;10^{2}$ +	9.54$\;\times \;10^{3}$ +	−9.95$\;\times \;10^{2}$ +	**−9.98$\;\times \;10^{2}$**
	std	4.96$\;\times \;10^{3}$	6.02$\;\times \;10^{4}$	3.08$\;\times \;10^{4}$	1.90$\;\times \;10^{3}$	6.90$\;\times \;10^{3}$	1.34$\;\times \;10^{2}$	3.43$\;\times \;10^{3}$	1.36$\;\times \;10^{0}$	**6.83$\;\times \;10^{-1}$**
$f_{6}$	mean	6.88$\;\times \;10^{3}$ +	5.05$\;\times \;10^{3}$ +	1.28$\;\times \;10^{4}$ +	−4.65$\;\times \;10^{2}$ +	6.72$\;\times \;10^{3}$ +	−7.86$\;\times \;10^{2}$ ≈	4.62$\;\times \;10^{2}$ +	−8.04$\;\times \;10^{2}$ ≈	**−8.11$\;\times \;10^{2}$**
	std	9.37$\;\times \;10^{2}$	2.49$\;\times \;10^{3}$	3.80$\;\times \;10^{3}$	1.17$\;\times \;10^{2}$	5.61$\;\times \;10^{2}$	5.00$\;\times \;10^{1}$	4.09$\;\times \;10^{2}$	4.49$\;\times \;10^{1}$	**4.28$\;\times \;10^{1}$**
$f_{7}$	mean	8.19$\;\times \;10^{5}$ +	2.86$\;\times \;10^{7}$ +	3.61$\;\times \;10^{8}$ +	4.97$\;\times \;10^{6}$ +	1.02$\;\times \;10^{7}$ +	**4.38$\;\times \;10^{5}$** ≈	5.85$\;\times \;10^{5}$ ≈	3.95$\;\times \;10^{6}$ +	4.69$\;\times \;10^{5}$
	std	7.36$\;\times \;10^{4}$	4.76$\;\times \;10^{7}$	3.59$\;\times \;10^{8}$	7.04$\;\times \;10^{6}$	3.79$\;\times \;10^{6}$	**6.13$\;\times \;10^{4}$**	2.37$\;\times \;10^{5}$	8.69$\;\times \;10^{6}$	8.88$\;\times \;10^{4}$
$f_{8}$	mean	−6.79$\;\times \;10^{2}$ ≈	−6.79$\;\times \;10^{2}$ ≈	−6.79$\;\times \;10^{2}$ ≈	−6.79$\;\times \;10^{2}$ ≈	−6.79$\;\times \;10^{2}$ ≈	**−6.79$\;\times \;10^{2}$** ≈	−6.79$\;\times \;10^{2}$ ≈	−6.79$\;\times \;10^{2}$ ≈	−6.79$\;\times \;10^{2}$
	std	3.54$\;\times \;10^{-2}$	3.53$\;\times \;10^{-2}$	4.84$\;\times \;10^{-2}$	6.62$\;\times \;10^{-2}$	3.84$\;\times \;10^{-2}$	**5.28$\;\times \;10^{-2}$**	5.53$\;\times \;10^{-2}$	4.25$\;\times \;10^{-2}$	3.67$\;\times \;10^{-2}$
$f_{9}$	mean	−5.24$\;\times \;10^{2}$ +	−5.30$\;\times \;10^{2}$ +	−5.22$\;\times \;10^{2}$ +	−5.28$\;\times \;10^{2}$ +	−5.23$\;\times \;10^{2}$ +	**−5.47$\;\times \;10^{2}$** −	−5.40$\;\times \;10^{2}$ +	−5.38$\;\times \;10^{2}$ +	−5.43$\;\times \;10^{2}$
	std	1.23$\;\times \;10^{0}$	4.42$\;\times \;10^{0}$	1.91$\;\times \;10^{0}$	3.59$\;\times \;10^{0}$	1.33$\;\times \;10^{0}$	**5.63$\;\times \;10^{0}$**	3.54$\;\times \;10^{0}$	4.62$\;\times \;10^{0}$	5.21$\;\times \;10^{0}$
$f_{10}$	mean	8.37$\;\times \;10^{3}$ +	9.74$\;\times \;10^{3}$ +	1.67$\;\times \;10^{4}$ +	5.27$\;\times \;10^{2}$ +	1.00$\;\times \;10^{4}$ +	−2.63$\;\times \;10^{2}$ +	2.51$\;\times \;10^{3}$ +	−4.88$\;\times \;10^{2}$ ≈	**−4.89$\;\times \;10^{2}$**
	std	1.32$\;\times \;10^{3}$	2.64$\;\times \;10^{3}$	2.67$\;\times \;10^{3}$	3.40$\;\times \;10^{2}$	6.53$\;\times \;10^{2}$	7.04$\;\times \;10^{1}$	7.06$\;\times \;10^{2}$	1.88$\;\times \;10^{0}$	**2.29$\;\times \;10^{0}$**
$f_{11}$	mean	9.78$\;\times \;10^{2}$ +	9.69$\;\times \;10^{2}$ +	1.08$\;\times \;10^{3}$ +	4.05$\;\times \;10^{2}$ +	7.17$\;\times \;10^{2}$ +	−7.11$\;\times \;10^{1}$ +	2.72$\;\times \;10^{2}$ +	5.33$\;\times \;10^{2}$ +	**−3.66$\;\times \;10^{2}$**
	std	7.25$\;\times \;10^{1}$	2.36$\;\times \;10^{2}$	2.16$\;\times \;10^{2}$	1.12$\;\times \;10^{2}$	4.65$\;\times \;10^{1}$	4.88$\;\times \;10^{1}$	7.64$\;\times \;10^{1}$	1.51$\;\times \;10^{2}$	**7.80$\;\times \;10^{0}$**
$f_{12}$	mean	1.10$\;\times \;10^{3}$ +	9.74$\;\times \;10^{2}$ +	1.31$\;\times \;10^{3}$ +	5.70$\;\times \;10^{2}$ +	8.89$\;\times \;10^{2}$ +	**8.24$\;\times \;10^{1}$** −	3.89$\;\times \;10^{2}$ ≈	5.69$\;\times \;10^{2}$ +	4.08$\;\times \;10^{2}$
	std	1.07$\;\times \;10^{2}$	2.32$\;\times \;10^{2}$	1.68$\;\times \;10^{2}$	1.08$\;\times \;10^{2}$	4.11$\;\times \;10^{1}$	**6.73$\;\times \;10^{1}$**	9.51$\;\times \;10^{1}$	1.55$\;\times \;10^{2}$	1.30$\;\times \;10^{2}$
$f_{13}$	mean	1.18$\;\times \;10^{3}$ +	1.02$\;\times \;10^{3}$ +	1.31$\;\times \;10^{3}$ +	7.92$\;\times \;10^{2}$ +	8.97$\;\times \;10^{2}$ +	**2.89$\;\times \;10^{2}$** −	5.05$\;\times \;10^{2}$ −	7.81$\;\times \;10^{2}$ +	5.80$\;\times \;10^{2}$
	std	1.08$\;\times \;10^{2}$	1.87$\;\times \;10^{2}$	1.61$\;\times \;10^{2}$	8.04$\;\times \;10^{1}$	3.70$\;\times \;10^{1}$	**7.01$\;\times \;10^{1}$**	8.49$\;\times \;10^{1}$	1.59$\;\times \;10^{2}$	1.35$\;\times \;10^{2}$
$f_{14}$	mean	1.12$\;\times \;10^{4}$ +	1.53$\;\times \;10^{4}$ +	1.49$\;\times \;10^{4}$ +	1.02$\;\times \;10^{4}$ +	1.52$\;\times \;10^{4}$ +	6.11$\;\times \;10^{3}$ +	1.08$\;\times \;10^{4}$ +	8.28$\;\times \;10^{3}$ +	**3.27$\;\times \;10^{2}$**
	std	4.80$\;\times \;10^{2}$	7.13$\;\times \;10^{2}$	5.68$\;\times \;10^{2}$	1.24$\;\times \;10^{3}$	4.71$\;\times \;10^{2}$	7.76$\;\times \;10^{2}$	1.21$\;\times \;10^{3}$	9.40$\;\times \;10^{2}$	**1.62$\;\times \;10^{2}$**
$f_{15}$	mean	1.50$\;\times \;10^{4}$ +	1.56$\;\times \;10^{4}$ +	1.58$\;\times \;10^{4}$ +	1.27$\;\times \;10^{4}$ +	1.51$\;\times \;10^{4}$ +	1.08$\;\times \;10^{4}$ +	1.23$\;\times \;10^{4}$ +	8.77$\;\times \;10^{3}$ ≈	**8.67$\;\times \;10^{3}$**
	std	3.75$\;\times \;10^{2}$	4.18$\;\times \;10^{2}$	3.84$\;\times \;10^{2}$	1.39$\;\times \;10^{3}$	5.10$\;\times \;10^{2}$	1.10$\;\times \;10^{3}$	9.49$\;\times \;10^{2}$	7.62$\;\times \;10^{2}$	**8.16$\;\times \;10^{2}$**
$f_{16}$	mean	2.04$\;\times \;10^{2}$ +	2.04$\;\times \;10^{2}$ +	2.04$\;\times \;10^{2}$ +	2.03$\;\times \;10^{2}$ +	2.04$\;\times \;10^{2}$ +	2.03$\;\times \;10^{2}$ +	**2.02$\;\times \;10^{2}$** ≈	2.03$\;\times \;10^{2}$ +	2.02$\;\times \;10^{2}$
	std	3.34$\;\times \;10^{-1}$	3.08$\;\times \;10^{-1}$	5.68$\;\times \;10^{-1}$	7.99$\;\times \;10^{-1}$	2.64$\;\times \;10^{-1}$	6.25$\;\times \;10^{-1}$	**4.24$\;\times \;10^{-1}$**	5.45$\;\times \;10^{-1}$	6.19$\;\times \;10^{-1}$
$f_{17}$	mean	4.36$\;\times \;10^{3}$ +	2.69$\;\times \;10^{3}$ +	2.34$\;\times \;10^{3}$ +	1.60$\;\times \;10^{3}$ +	1.69$\;\times \;10^{3}$ +	8.28$\;\times \;10^{2}$ +	1.19$\;\times \;10^{3}$ +	2.12$\;\times \;10^{3}$ +	**4.09$\;\times \;10^{2}$**
	std	4.57$\;\times \;10^{2}$	5.31$\;\times \;10^{2}$	2.83$\;\times \;10^{2}$	1.54$\;\times \;10^{2}$	2.83$\;\times \;10^{1}$	6.32$\;\times \;10^{1}$	1.05$\;\times \;10^{2}$	3.14$\;\times \;10^{2}$	**9.50$\;\times \;10^{0}$**
$f_{18}$	mean	3.99$\;\times \;10^{3}$ +	4.00$\;\times \;10^{3}$ +	4.75$\;\times \;10^{3}$ +	3.02$\;\times \;10^{3}$ +	3.78$\;\times \;10^{3}$ +	**9.54$\;\times \;10^{2}$** ≈	2.00$\;\times \;10^{3}$ +	3.45$\;\times \;10^{3}$ +	1.02$\;\times \;10^{3}$
	std	2.33$\;\times \;10^{2}$	6.43$\;\times \;10^{2}$	4.75$\;\times \;10^{2}$	2.93$\;\times \;10^{2}$	1.14$\;\times \;10^{2}$	**8.31$\;\times \;10^{1}$**	2.36$\;\times \;10^{2}$	5.13$\;\times \;10^{2}$	1.25$\;\times \;10^{2}$
$f_{19}$	mean	2.71$\;\times \;10^{6}$ +	6.27$\;\times \;10^{5}$ +	2.07$\;\times \;10^{6}$ +	4.33$\;\times \;10^{3}$ +	3.27$\;\times \;10^{5}$ +	5.56$\;\times \;10^{2}$ +	1.59$\;\times \;10^{4}$ +	5.68$\;\times \;10^{2}$ +	**5.39$\;\times \;10^{2}$**
	std	8.77$\;\times \;10^{5}$	7.00$\;\times \;10^{5}$	1.62$\;\times \;10^{6}$	3.63$\;\times \;10^{3}$	5.09$\;\times \;10^{4}$	9.34$\;\times \;10^{0}$	1.08$\;\times \;10^{4}$	1.10$\;\times \;10^{1}$	**6.63$\;\times \;10^{0}$**
$f_{20}$	mean	6.24$\;\times \;10^{2}$ ≈	6.25$\;\times \;10^{2}$ +	6.25$\;\times \;10^{2}$ +	6.25$\;\times \;10^{2}$ +	6.25$\;\times \;10^{2}$ +	**6.24$\;\times \;10^{2}$** ≈	6.24$\;\times \;10^{2}$ ≈	6.24$\;\times \;10^{2}$ ≈	6.24$\;\times \;10^{2}$
	std	2.35$\;\times \;10^{-1}$	2.20$\;\times \;10^{-1}$	1.91$\;\times \;10^{-6}$	7.82$\;\times \;10^{-2}$	6.78$\;\times \;10^{-2}$	**6.35$\;\times \;10^{-1}$**	3.66$\;\times \;10^{-1}$	3.66$\;\times \;10^{-1}$	5.46$\;\times \;10^{-1}$
$f_{21}$	mean	7.93$\;\times \;10^{3}$ +	6.90$\;\times \;10^{3}$ +	6.36$\;\times \;10^{3}$ +	1.78$\;\times \;10^{3}$ +	4.58$\;\times \;10^{3}$ +	1.36$\;\times \;10^{3}$ +	3.48$\;\times \;10^{3}$ +	1.19$\;\times \;10^{3}$ +	**1.14$\;\times \;10^{3}$**
	std	5.48$\;\times \;10^{2}$	1.43$\;\times \;10^{3}$	8.57$\;\times \;10^{2}$	3.50$\;\times \;10^{2}$	9.06$\;\times \;10^{1}$	4.11$\;\times \;10^{2}$	2.71$\;\times \;10^{2}$	2.17$\;\times \;10^{2}$	**3.17$\;\times \;10^{0}$**
$f_{22}$	mean	1.29$\;\times \;10^{4}$ +	1.73$\;\times \;10^{4}$ +	1.72$\;\times \;10^{4}$ +	1.30$\;\times \;10^{4}$ +	1.75$\;\times \;10^{4}$ +	7.88$\;\times \;10^{3}$ +	1.43$\;\times \;10^{4}$ +	1.12$\;\times \;10^{4}$ +	**1.31$\;\times \;10^{3}$**
	std	5.04$\;\times \;10^{2}$	6.23$\;\times \;10^{2}$	4.88$\;\times \;10^{2}$	1.41$\;\times \;10^{3}$	3.98$\;\times \;10^{2}$	9.87$\;\times \;10^{2}$	1.43$\;\times \;10^{3}$	1.17$\;\times \;10^{3}$	**1.33$\;\times \;10^{2}$**
$f_{23}$	mean	1.61$\;\times \;10^{4}$ +	1.67$\;\times \;10^{4}$ +	1.73$\;\times \;10^{4}$ +	1.48$\;\times \;10^{4}$ +	1.71$\;\times \;10^{4}$ +	1.23$\;\times \;10^{4}$ +	1.50$\;\times \;10^{4}$ +	1.11$\;\times \;10^{4}$ ≈	**1.09$\;\times \;10^{4}$**
	std	3.57$\;\times \;10^{2}$	7.21$\;\times \;10^{2}$	4.83$\;\times \;10^{2}$	1.21$\;\times \;10^{3}$	4.20$\;\times \;10^{2}$	1.17$\;\times \;10^{3}$	1.42$\;\times \;10^{3}$	1.06$\;\times \;10^{3}$	**9.74$\;\times \;10^{2}$**
$f_{24}$	mean	1.40$\;\times \;10^{3}$ +	1.41$\;\times \;10^{3}$ +	1.67$\;\times \;10^{3}$ +	1.44$\;\times \;10^{3}$ +	1.62$\;\times \;10^{3}$ +	**1.33$\;\times \;10^{3}$** −	1.41$\;\times \;10^{3}$ +	1.41$\;\times \;10^{3}$ +	1.36$\;\times \;10^{3}$
	std	5.83$\;\times \;10^{0}$	1.68$\;\times \;10^{1}$	1.38$\;\times \;10^{2}$	3.20$\;\times \;10^{1}$	7.81$\;\times \;10^{1}$	**1.57$\;\times \;10^{1}$**	2.44$\;\times \;10^{1}$	1.29$\;\times \;10^{1}$	1.47$\;\times \;10^{1}$
$f_{25}$	mean	1.51$\;\times \;10^{3}$ ≈	1.52$\;\times \;10^{3}$ +	1.59$\;\times \;10^{3}$ +	1.54$\;\times \;10^{3}$ +	1.62$\;\times \;10^{3}$ +	**1.46$\;\times \;10^{3}$** −	1.53$\;\times \;10^{3}$ +	1.58$\;\times \;10^{3}$ +	1.50$\;\times \;10^{3}$
	std	8.24$\;\times \;10^{0}$	1.63$\;\times \;10^{1}$	2.61$\;\times \;10^{1}$	1.66$\;\times \;10^{1}$	2.50$\;\times \;10^{1}$	**1.44$\;\times \;10^{1}$**	1.93$\;\times \;10^{1}$	2.38$\;\times \;10^{1}$	1.84$\;\times \;10^{1}$
$f_{26}$	mean	1.69$\;\times \;10^{3}$ +	1.68$\;\times \;10^{3}$ +	1.74$\;\times \;10^{3}$ +	1.70$\;\times \;10^{3}$ +	1.73$\;\times \;10^{3}$ +	**1.62$\;\times \;10^{3}$** −	1.64$\;\times \;10^{3}$ +	1.67$\;\times \;10^{3}$ +	1.64$\;\times \;10^{3}$
	std	5.49$\;\times \;10^{0}$	1.17$\;\times \;10^{1}$	4.18$\;\times \;10^{1}$	1.41$\;\times \;10^{1}$	7.34$\;\times \;10^{0}$	**4.17$\;\times \;10^{1}$**	6.64$\;\times \;10^{1}$	1.30$\;\times \;10^{1}$	1.10$\;\times \;10^{1}$
$f_{27}$	mean	3.63$\;\times \;10^{3}$ +	3.64$\;\times \;10^{3}$ +	4.77$\;\times \;10^{3}$ +	3.82$\;\times \;10^{3}$ +	4.52$\;\times \;10^{3}$ +	**2.86$\;\times \;10^{3}$** −	3.47$\;\times \;10^{3}$ +	3.57$\;\times \;10^{3}$ +	3.25$\;\times \;10^{3}$
	std	5.36$\;\times \;10^{1}$	1.31$\;\times \;10^{2}$	5.24$\;\times \;10^{2}$	2.13$\;\times \;10^{2}$	1.93$\;\times \;10^{2}$	**1.35$\;\times \;10^{2}$**	1.77$\;\times \;10^{2}$	2.51$\;\times \;10^{2}$	1.28$\;\times \;10^{2}$
$f_{28}$	mean	9.18$\;\times \;10^{3}$ +	1.27$\;\times \;10^{4}$ +	1.33$\;\times \;10^{4}$ +	9.65$\;\times \;10^{3}$ +	9.95$\;\times \;10^{3}$ +	**3.21$\;\times \;10^{3}$** ≈	7.66$\;\times \;10^{3}$ +	8.88$\;\times \;10^{3}$ +	4.93$\;\times \;10^{3}$
	std	4.78$\;\times \;10^{2}$	1.74$\;\times \;10^{3}$	1.57$\;\times \;10^{3}$	9.01$\;\times \;10^{2}$	3.26$\;\times \;10^{2}$	**9.33$\;\times \;10^{2}$**	5.01$\;\times \;10^{2}$	1.16$\;\times \;10^{3}$	2.08$\;\times \;10^{3}$
		+/≈/−: 25/3/0	+/≈/−: 27/1/0	+/≈/−: 27/1/0	+/≈/−: 27/1/0	+/≈/−: 27/1/0	+/≈/−: 15/6/7	+/≈/−:v 22/5/1	+/≈/−: 20/6/2	

**Table 6 biomimetics-08-00454-t006:** Experimental and statistical results on seven engineering problems.

Func.		DE	PSO	DE-SAP	PPSO	SSDO	RIME	SAO	VEGE	Proposal
WBP	mean	2.0480$\;\times \;10^{0}$ +	2.0480$\;\times \;10^{0}$ +	2.3833$\;\times \;10^{0}$ +	2.2624$\;\times \;10^{0}$ +	1.4229$\;\times \;10^{5}$ +	1.9252$\;\times \;10^{0}$ +	1.9104$\;\times \;10^{0}$ +	1.7238$\;\times \;10^{0}$ +	**1.6916$\;\times \;10^{0}$**
	std	2.0936$\;\times \;10^{-1}$	2.0936$\;\times \;10^{-1}$	3.4810$\;\times \;10^{-1}$	5.3722$\;\times \;10^{-1}$	7.6623$\;\times \;10^{5}$	2.2225$\;\times \;10^{-1}$	2.5516$\;\times \;10^{-1}$	**3.1670$\;\times \;10^{-2}$**	4.3967$\;\times \;10^{-2}$
	worst	2.5772$\;\times \;10^{0}$	2.5772$\;\times \;10^{0}$	3.1889$\;\times \;10^{0}$	3.8419$\;\times \;10^{0}$	4.2686$\;\times \;10^{6}$	2.8182$\;\times \;10^{0}$	2.8941$\;\times \;10^{0}$	**1.8171$\;\times \;10^{0}$**	1.9271$\;\times \;10^{0}$
	best	1.7887$\;\times \;10^{0}$	1.7887$\;\times \;10^{0}$	1.8893$\;\times \;10^{0}$	1.7229$\;\times \;10^{0}$	2.6867$\;\times \;10^{0}$	1.7036$\;\times \;10^{0}$	1.6829$\;\times \;10^{0}$	1.6848$\;\times \;10^{0}$	**1.6826$\;\times \;10^{0}$**
CSP	mean	6.0775$\;\times \;10^{-3}$ +	6.0775$\;\times \;10^{-3}$ +	6.5158$\;\times \;10^{-3}$ +	6.0990$\;\times \;10^{-3}$ ≈	6.0818$\;\times \;10^{-3}$ +	6.0785$\;\times \;10^{-3}$ +	6.1013$\;\times \;10^{-3}$ +	6.0871$\;\times \;10^{-3}$ +	**6.0761$\;\times \;10^{-3}$**
	std	1.5168$\;\times \;10^{-6}$	1.5168$\;\times \;10^{-6}$	5.2997$\;\times \;10^{-4}$	1.2300$\;\times \;10^{-4}$	5.9274$\;\times \;10^{-6}$	4.0493$\;\times \;10^{-6}$	1.2203$\;\times \;10^{-4}$	1.2788$\;\times \;10^{-5}$	**1.3320$\;\times \;10^{-8}$**
	worst	6.0811$\;\times \;10^{-3}$	6.0811$\;\times \;10^{-3}$	8.5403$\;\times \;10^{-3}$	6.7614$\;\times \;10^{-3}$	6.0986$\;\times \;10^{-3}$	6.0978$\;\times \;10^{-3}$	6.7582$\;\times \;10^{-3}$	6.1218$\;\times \;10^{-3}$	**6.0762$\;\times \;10^{-3}$**
	best	**6.0761$\;\times \;10^{-3}$**	**6.0761$\;\times \;10^{-3}$**	6.0777$\;\times \;10^{-3}$	**6.0761$\;\times \;10^{-3}$**	6.0763$\;\times \;10^{-3}$	6.0762$\;\times \;10^{-3}$	**6.0761$\;\times \;10^{-3}$**	**6.0761$\;\times \;10^{-3}$**	**6.0761$\;\times \;10^{-3}$**
SRD	mean	3.0695$\;\times \;10^{3}$ +	3.0695$\;\times \;10^{3}$ +	3.2475$\;\times \;10^{3}$ +	3.0741$\;\times \;10^{3}$ +	1.2766$\;\times \;10^{6}$ +	2.9963$\;\times \;10^{3}$ +	2.9990$\;\times \;10^{3}$ +	3.0450$\;\times \;10^{3}$ +	**2.9869$\;\times \;10^{3}$**
	std	5.3510$\;\times \;10^{1}$	5.3510$\;\times \;10^{1}$	4.3277$\;\times \;10^{1}$	9.4483$\;\times \;10^{1}$	1.7862$\;\times \;10^{6}$	7.0867$\;\times \;10^{0}$	9.1124$\;\times \;10^{0}$	2.7661$\;\times \;10^{1}$	**1.0080$\;\times \;10^{-3}$**
	worst	3.2223$\;\times \;10^{3}$	3.2223$\;\times \;10^{3}$	3.3639$\;\times \;10^{3}$	3.3639$\;\times \;10^{3}$	6.1677$\;\times \;10^{6}$	3.0180$\;\times \;10^{3}$	3.0377$\;\times \;10^{3}$	3.1019$\;\times \;10^{3}$	**2.9869$\;\times \;10^{3}$**
	best	3.0298$\;\times \;10^{3}$	3.0298$\;\times \;10^{3}$	3.1683$\;\times \;10^{3}$	2.9879$\;\times \;10^{3}$	3.2053$\;\times \;10^{3}$	2.9876$\;\times \;10^{3}$	2.9886$\;\times \;10^{3}$	3.0000$\;\times \;10^{3}$	**2.9869$\;\times \;10^{3}$**
TBTD	mean	2.6419$\;\times \;10^{2}$ +	2.6419$\;\times \;10^{2}$ +	2.6864$\;\times \;10^{2}$ +	2.6413$\;\times \;10^{2}$ +	2.6999$\;\times \;10^{2}$ +	2.6413$\;\times \;10^{2}$ +	2.6423$\;\times \;10^{2}$ +	**2.6390$\;\times \;10^{2}$** ≈	2.6390$\;\times \;10^{2}$
	std	1.8476$\;\times \;10^{-1}$	1.8476$\;\times \;10^{-1}$	2.0998$\;\times \;10^{0}$	3.8377$\;\times \;10^{-1}$	5.0347$\;\times \;10^{0}$	3.8677$\;\times \;10^{-1}$	1.1078$\;\times \;10^{0}$	**2.4545$\;\times \;10^{-5}$**	2.2217$\;\times \;10^{-3}$
	worst	2.6475$\;\times \;10^{2}$	2.6475$\;\times \;10^{2}$	2.7107$\;\times \;10^{2}$	2.6522$\;\times \;10^{2}$	2.8284$\;\times \;10^{2}$	2.6553$\;\times \;10^{2}$	2.6985$\;\times \;10^{2}$	**2.6390$\;\times \;10^{2}$**	2.6391$\;\times \;10^{2}$
	best	2.6395$\;\times \;10^{2}$	2.6395$\;\times \;10^{2}$	2.6419$\;\times \;10^{2}$	**2.6390$\;\times \;10^{2}$**	2.6400$\;\times \;10^{2}$	**2.6390$\;\times \;10^{2}$**	**2.6390$\;\times \;10^{2}$**	**2.6390$\;\times \;10^{2}$**	**2.6390$\;\times \;10^{2}$**
CBD	mean	2.7914$\;\times \;10^{0}$ +	2.7914$\;\times \;10^{0}$ +	4.0830$\;\times \;10^{0}$ +	2.8157$\;\times \;10^{0}$ +	2.5804$\;\times \;10^{0}$ +	1.3574$\;\times \;10^{0}$ +	1.3415$\;\times \;10^{0}$ +	1.3432$\;\times \;10^{0}$ +	**1.3402$\;\times \;10^{0}$**
	std	5.7027$\;\times \;10^{-1}$	5.7027$\;\times \;10^{-1}$	1.8919$\;\times \;10^{0}$	1.1259$\;\times \;10^{0}$	6.3506$\;\times \;10^{-1}$	1.5191$\;\times \;10^{-2}$	1.7858$\;\times \;10^{-3}$	1.8963$\;\times \;10^{-3}$	**4.4879$\;\times \;10^{-4}$**
	worst	4.2211$\;\times \;10^{0}$	4.2211$\;\times \;10^{0}$	8.3554$\;\times \;10^{0}$	5.5371$\;\times \;10^{0}$	3.8432$\;\times \;10^{0}$	1.4100$\;\times \;10^{0}$	1.3480$\;\times \;10^{0}$	1.3474$\;\times \;10^{0}$	**1.3423$\;\times \;10^{0}$**
	best	2.1447$\;\times \;10^{0}$	2.1447$\;\times \;10^{0}$	1.5797$\;\times \;10^{0}$	1.4229$\;\times \;10^{0}$	1.3684$\;\times \;10^{0}$	1.3409$\;\times \;10^{0}$	1.3401$\;\times \;10^{0}$	1.3405$\;\times \;10^{0}$	**1.3400$\;\times \;10^{0}$**
TCD	mean	3.0542$\;\times \;10^{1}$ +	3.0542$\;\times \;10^{1}$ +	3.0904$\;\times \;10^{1}$ +	3.0254$\;\times \;10^{1}$ +	3.1549$\;\times \;10^{1}$ +	3.0328$\;\times \;10^{1}$ +	3.0153$\;\times \;10^{1}$ +	**3.0150$\;\times \;10^{1}$** ≈	3.0152$\;\times \;10^{1}$
	std	2.3954$\;\times \;10^{-1}$	2.3954$\;\times \;10^{-1}$	4.4253$\;\times \;10^{-1}$	3.8837$\;\times \;10^{-1}$	6.3834$\;\times \;10^{-1}$	2.0913$\;\times \;10^{-1}$	1.0563$\;\times \;10^{-2}$	**5.3976$\;\times \;10^{-4}$**	1.1610$\;\times \;10^{-2}$
	worst	3.1181$\;\times \;10^{1}$	3.1181$\;\times \;10^{1}$	3.2113$\;\times \;10^{1}$	3.2213$\;\times \;10^{1}$	3.2711$\;\times \;10^{1}$	3.0857$\;\times \;10^{1}$	3.0208$\;\times \;10^{1}$	**3.0153$\;\times \;10^{1}$**	3.0214$\;\times \;10^{1}$
	best	3.0192$\;\times \;10^{1}$	3.0192$\;\times \;10^{1}$	3.0287$\;\times \;10^{1}$	**3.0150$\;\times \;10^{1}$**	3.0289$\;\times \;10^{1}$	3.0151$\;\times \;10^{1}$	**3.0150$\;\times \;10^{1}$**	**3.0150$\;\times \;10^{1}$**	**3.0150$\;\times \;10^{1}$**
CBHD	mean	7.8378$\;\times \;10^{0}$ +	7.8378$\;\times \;10^{0}$ +	8.5143$\;\times \;10^{0}$ +	7.2506$\;\times \;10^{0}$ +	1.0699$\;\times \;10^{1}$ +	6.8542$\;\times \;10^{0}$ +	6.9393$\;\times \;10^{0}$ +	6.8762$\;\times \;10^{0}$ +	**6.8430$\;\times \;10^{0}$**
	std	7.1741$\;\times \;10^{-1}$	7.1741$\;\times \;10^{-1}$	6.6499$\;\times \;10^{-1}$	4.0215$\;\times \;10^{-1}$	1.2514$\;\times \;10^{0}$	7.7909$\;\times \;10^{-3}$	1.5303$\;\times \;10^{-1}$	4.5820$\;\times \;10^{-2}$	**3.0522$\;\times \;10^{-8}$**
	worst	1.0352$\;\times \;10^{1}$	1.0352$\;\times \;10^{1}$	1.0069$\;\times \;10^{1}$	8.4715$\;\times \;10^{0}$	1.3665$\;\times \;10^{1}$	6.8709$\;\times \;10^{0}$	7.5109$\;\times \;10^{0}$	7.0912$\;\times \;10^{0}$	**6.8430$\;\times \;10^{0}$**
	best	7.0410$\;\times \;10^{0}$	7.0410$\;\times \;10^{0}$	7.7228$\;\times \;10^{0}$	6.8774$\;\times \;10^{0}$	8.2201$\;\times \;10^{0}$	6.8444$\;\times \;10^{0}$	6.8431$\;\times \;10^{0}$	6.8451$\;\times \;10^{0}$	**6.8430$\;\times \;10^{0}$**
		+/≈/−: 7/0/0	+/≈/−: 7/0/0	+/≈/−: 7/0/0	+/≈/−: 6/1/0	+/≈/−: 7/0/0	+/≈/−: 7/0/0	+/≈/−: 7/0/0	+/≈/−: 5/2/0	

**Table 7 biomimetics-08-00454-t007:** Ablation experimental results on 30-D CEC2013 benchmark functions.

Func	VEGE		VEGE-i		VEGE-ii		Proposal	
	Mean	Std	Mean	Std	Mean	Std	Mean	Std
$f_{1}$	−1.40$\;\times \;10^{3}$ +	8.37$\;\times \;10^{-1}$	−1.40$\;\times \;10^{3}$ +	6.88$\;\times \;10^{-1}$	**−1.40$\;\times \;10^{3}$** ≈	**1.32$\;\times \;10^{-1}$**	−1.40$\;\times \;10^{3}$	1.63$\;\times \;10^{-1}$
$f_{2}$	5.43$\;\times \;10^{6}$ ≈	2.36$\;\times \;10^{6}$	**4.84$\;\times \;10^{6}$** ≈	**2.22$\;\times \;10^{6}$**	4.90$\;\times \;10^{6}$ ≈	1.92$\;\times \;10^{6}$	5.35$\;\times \;10^{6}$	2.50$\;\times \;10^{6}$
$f_{3}$	2.98$\;\times \;10^{7}$ +	4.89$\;\times \;10^{7}$	6.16$\;\times \;10^{7}$ +	1.89$\;\times \;10^{8}$	**4.43$\;\times \;10^{6}$** ≈	**8.65$\;\times \;10^{6}$**	5.73$\;\times \;10^{6}$	9.05$\;\times \;10^{6}$
$f_{4}$	1.54$\;\times \;10^{4}$ ≈	6.52$\;\times \;10^{3}$	1.47$\;\times \;10^{4}$ ≈	6.42$\;\times \;10^{3}$	1.44$\;\times \;10^{4}$ ≈	6.29$\;\times \;10^{3}$	**1.39$\;\times \;10^{4}$**	**5.15$\;\times \;10^{3}$**
$f_{5}$	−9.97$\;\times \;10^{2}$ +	4.32$\;\times \;10^{0}$	−9.97$\;\times \;10^{2}$ +	2.62$\;\times \;10^{0}$	−9.99$\;\times \;10^{2}$ ≈	2.55$\;\times \;10^{-1}$	**−9.99$\;\times \;10^{2}$**	**2.07$\;\times \;10^{-1}$**
$f_{6}$	−8.32$\;\times \;10^{2}$ ≈	3.04$\;\times \;10^{1}$	−8.36$\;\times \;10^{2}$ ≈	3.15$\;\times \;10^{1}$	−8.39$\;\times \;10^{2}$ ≈	2.99$\;\times \;10^{1}$	**−8.40$\;\times \;10^{2}$**	**3.62$\;\times \;10^{1}$**
$f_{7}$	3.42$\;\times \;10^{6}$ +	1.24$\;\times \;10^{7}$	2.45$\;\times \;10^{6}$ +	8.68$\;\times \;10^{6}$	1.40$\;\times \;10^{5}$ ≈	2.95$\;\times \;10^{4}$	**1.40$\;\times \;10^{5}$**	**4.08$\;\times \;10^{4}$**
$f_{8}$	−6.79$\;\times \;10^{2}$ ≈	4.44$\;\times \;10^{-2}$	−6.79$\;\times \;10^{2}$ ≈	4.98$\;\times \;10^{-2}$	**−6.79$\;\times \;10^{2}$** ≈	**6.07$\;\times \;10^{-2}$**	−6.79$\;\times \;10^{2}$	4.68$\;\times \;10^{-2}$
$f_{9}$	−5.68$\;\times \;10^{2}$ ≈	3.98$\;\times \;10^{0}$	−5.67$\;\times \;10^{2}$ +	3.50$\;\times \;10^{0}$	−**5.73$\;\times \;10^{2}$** ≈	**3.57$\;\times \;10^{0}$**	−5.71$\;\times \;10^{2}$	4.13$\;\times \;10^{0}$
$f_{10}$	−4.96$\;\times \;10^{2}$ ≈	6.48$\;\times \;10^{-1}$	−4.96$\;\times \;10^{2}$ +	7.67$\;\times \;10^{-1}$	−4.97$\;\times \;10^{2}$ ≈	9.79$\;\times \;10^{-1}$	**−4.97$\;\times \;10^{2}$**	**6.01$\;\times \;10^{-1}$**
$f_{11}$	1.03$\;\times \;10^{2}$ +	1.03$\;\times \;10^{2}$	6.81$\;\times \;10^{1}$ +	1.12$\;\times \;10^{2}$	−3.83$\;\times \;10^{2}$ ≈	5.52$\;\times \;10^{0}$	**−3.84$\;\times \;10^{2}$**	**4.27$\;\times \;10^{0}$**
$f_{12}$	1.54$\;\times \;10^{2}$ +	1.10$\;\times \;10^{2}$	1.95$\;\times \;10^{2}$ +	8.51$\;\times \;10^{1}$	**3.07$\;\times \;10^{1}$** ≈	**1.06$\;\times \;10^{2}$**	5.69$\;\times \;10^{1}$	9.84$\;\times \;10^{1}$
$f_{13}$	3.70$\;\times \;10^{2}$ +	1.24$\;\times \;10^{2}$	3.91$\;\times \;10^{2}$ +	9.34$\;\times \;10^{1}$	1.98$\;\times \;10^{2}$ ≈	8.52$\;\times \;10^{1}$	**1.91$\;\times \;10^{2}$**	**7.60$\;\times \;10^{1}$**
$f_{14}$	4.59$\;\times \;10^{3}$ +	4.93$\;\times \;10^{2}$	4.52$\;\times \;10^{3}$ +	5.85$\;\times \;10^{2}$	**2.39$\;\times \;10^{2}$** ≈	**1.21$\;\times \;10^{2}$**	2.49$\;\times \;10^{2}$	1.66$\;\times \;10^{2}$
$f_{15}$	4.52$\;\times \;10^{3}$ ≈	4.96$\;\times \;10^{2}$	4.55$\;\times \;10^{3}$ ≈	5.89$\;\times \;10^{2}$	4.23$\;\times \;10^{3}$ ≈	5.42$\;\times \;10^{2}$	**4.20$\;\times \;10^{3}$**	**5.88$\;\times \;10^{2}$**
$f_{16}$	2.02$\;\times \;10^{2}$ +	4.48$\;\times \;10^{-1}$	2.02$\;\times \;10^{2}$ +	4.48$\;\times \;10^{-1}$	**2.01$\;\times \;10^{2}$** ≈	**6.01$\;\times \;10^{-1}$**	2.01$\;\times \;10^{2}$	3.57$\;\times \;10^{-1}$
$f_{17}$	1.15$\;\times \;10^{3}$ +	1.94$\;\times \;10^{2}$	1.22$\;\times \;10^{3}$ +	1.42$\;\times \;10^{2}$	**3.51$\;\times \;10^{2}$** ≈	**5.10$\;\times \;10^{0}$**	3.54$\;\times \;10^{2}$	5.91$\;\times \;10^{0}$
$f_{18}$	1.92$\;\times \;10^{3}$ +	3.64$\;\times \;10^{2}$	1.93$\;\times \;10^{3}$ +	3.67$\;\times \;10^{2}$	7.10$\;\times \;10^{2}$ ≈	8.06$\;\times \;10^{1}$	**7.08$\;\times \;10^{2}$**	**8.71$\;\times \;10^{1}$**
$f_{19}$	5.32$\;\times \;10^{2}$ +	5.45$\;\times \;10^{0}$	5.33$\;\times \;10^{2}$ +	4.19$\;\times \;10^{0}$	**5.19$\;\times \;10^{2}$** ≈	**4.93$\;\times \;10^{0}$**	5.19$\;\times \;10^{2}$	6.41$\;\times \;10^{0}$
$f_{20}$	6.15$\;\times \;10^{2}$ +	1.96$\;\times \;10^{-1}$	6.14$\;\times \;10^{2}$ ≈	1.30$\;\times \;10^{-1}$	6.14$\;\times \;10^{2}$ ≈	6.85$\;\times \;10^{-1}$	**6.14$\;\times \;10^{2}$**	**7.55$\;\times \;10^{-1}$**
$f_{21}$	1.83$\;\times \;10^{3}$ +	2.45$\;\times \;10^{2}$	1.75$\;\times \;10^{3}$ +	2.96$\;\times \;10^{2}$	1.75$\;\times \;10^{3}$ ≈	2.16$\;\times \;10^{2}$	**1.69$\;\times \;10^{3}$**	**2.69$\;\times \;10^{2}$**
$f_{22}$	6.26$\;\times \;10^{3}$ +	7.09$\;\times \;10^{2}$	6.30$\;\times \;10^{3}$ +	7.92$\;\times \;10^{2}$	**1.21$\;\times \;10^{3}$** ≈	**1.82$\;\times \;10^{2}$**	1.21$\;\times \;10^{3}$	1.40$\;\times \;10^{2}$
$f_{23}$	5.86$\;\times \;10^{3}$ ≈	6.87$\;\times \;10^{2}$	6.03$\;\times \;10^{3}$ ≈	6.28$\;\times \;10^{2}$	**5.70$\;\times \;10^{3}$** ≈	**6.43$\;\times \;10^{2}$**	5.85$\;\times \;10^{3}$	7.33$\;\times \;10^{2}$
$f_{24}$	1.30$\;\times \;10^{3}$ +	1.34$\;\times \;10^{1}$	1.30$\;\times \;10^{3}$ +	1.22$\;\times \;10^{1}$	1.28$\;\times \;10^{3}$ ≈	9.28$\;\times \;10^{0}$	**1.28$\;\times \;10^{3}$**	**1.12$\;\times \;10^{1}$**
$f_{25}$	1.43$\;\times \;10^{3}$ +	1.42$\;\times \;10^{1}$	1.44$\;\times \;10^{3}$ +	1.72$\;\times \;10^{1}$	**1.40$\;\times \;10^{3}$** ≈	**1.29$\;\times \;10^{1}$**	1.40$\;\times \;10^{3}$	1.62$\;\times \;10^{1}$
$f_{26}$	1.53$\;\times \;10^{3}$ +	8.66$\;\times \;10^{1}$	1.54$\;\times \;10^{3}$ +	8.37$\;\times \;10^{1}$	1.52$\;\times \;10^{3}$ ≈	8.05$\;\times \;10^{1}$	**1.51$\;\times \;10^{3}$**	**8.33$\;\times \;10^{1}$**
$f_{27}$	2.67$\;\times \;10^{3}$ +	1.12$\;\times \;10^{2}$	2.74$\;\times \;10^{3}$ +	1.55$\;\times \;10^{2}$	2.43$\;\times \;10^{3}$ ≈	1.24$\;\times \;10^{2}$	**2.43$\;\times \;10^{3}$**	**8.71$\;\times \;10^{1}$**
$f_{28}$	4.93$\;\times \;10^{3}$ +	4.48$\;\times \;10^{2}$	4.72$\;\times \;10^{3}$ +	6.73$\;\times \;10^{2}$	**3.57$\;\times \;10^{3}$** ≈	**1.03$\;\times \;10^{3}$**	3.64$\;\times \;10^{3}$	1.22$\;\times \;10^{3}$
	+/≈/−:20/8/0		+/≈/−:21/7/0		+/≈/−:0/28/0			

**Table 8 biomimetics-08-00454-t008:** Ablation experimental results on 50-D CEC2013 benchmark functions.

Func	VEGE		VEGE-i		VEGE-ii		Proposal	
	Mean	Std	Mean	Std	Mean	Std	Mean	Std
$f_{1}$	−1.39$\;\times \;10^{3}$ +	2.03$\;\times \;10^{0}$	−1.39$\;\times \;10^{3}$ +	1.79$\;\times \;10^{0}$	**−1.40$\;\times \;10^{3}$** ≈	**8.74$\;\times \;10^{-1}$**	−1.40$\;\times \;10^{3}$	1.07$\;\times \;10^{0}$
$f_{2}$	9.29$\;\times \;10^{6}$ −	2.54$\;\times \;10^{6}$	**8.94$\;\times \;10^{6}$** −	**2.44$\;\times \;10^{6}$**	1.02$\;\times \;10^{7}$ ≈	2.72$\;\times \;10^{6}$	1.15$\;\times \;10^{7}$	2.89$\;\times \;10^{6}$
$f_{3}$	2.21$\;\times \;10^{8}$ +	4.31$\;\times \;10^{8}$	2.83$\;\times \;10^{8}$ +	9.82$\;\times \;10^{8}$	7.89$\;\times \;10^{7}$ ≈	1.36$\;\times \;10^{8}$	**7.26$\;\times \;10^{7}$**	**1.70$\;\times \;10^{8}$**
$f_{4}$	**1.06$\;\times \;10^{4}$** −	**4.20$\;\times \;10^{3}$**	1.08$\;\times \;10^{4}$ −	3.81$\;\times \;10^{3}$	1.52$\;\times \;10^{4}$ ≈	6.96$\;\times \;10^{3}$	1.47$\;\times \;10^{4}$	4.77$\;\times \;10^{3}$
$f_{5}$	−9.95$\;\times \;10^{2}$ +	1.36$\;\times \;10^{0}$	−9.95$\;\times \;10^{2}$ +	1.13$\;\times \;10^{0}$	**−9.98$\;\times \;10^{2}$** ≈	**5.06$\;\times \;10^{-1}$**	−9.98$\;\times \;10^{2}$	6.83$\;\times \;10^{-1}$
$f_{6}$	−8.04$\;\times \;10^{2}$ ≈	4.49$\;\times \;10^{1}$	**−8.15$\;\times \;10^{2}$** ≈	**3.37$\;\times \;10^{1}$**	−8.09$\;\times \;10^{2}$ ≈	3.48$\;\times \;10^{1}$	−8.11$\;\times \;10^{2}$	4.28$\;\times \;10^{1}$
$f_{7}$	3.95$\;\times \;10^{6}$ +	8.69$\;\times \;10^{6}$	1.45$\;\times \;10^{6}$ +	1.38$\;\times \;10^{6}$	5.28$\;\times \;10^{5}$ ≈	2.43$\;\times \;10^{5}$	**4.69$\;\times \;10^{5}$**	**8.88$\;\times \;10^{4}$**
$f_{8}$	−6.79$\;\times \;10^{2}$ ≈	4.25$\;\times \;10^{-2}$	−6.79$\;\times \;10^{2}$ ≈	3.46$\;\times \;10^{-2}$	−6.79$\;\times \;10^{2}$ ≈	4.17$\;\times \;10^{-2}$	**−6.79$\;\times \;10^{2}$**	**3.67$\;\times \;10^{-2}$**
$f_{9}$	−5.38$\;\times \;10^{2}$ +	4.62$\;\times \;10^{0}$	−5.38$\;\times \;10^{2}$ +	4.32$\;\times \;10^{0}$	**−5.45$\;\times \;10^{2}$** ≈	**4.91$\;\times \;10^{0}$**	−5.43$\;\times \;10^{2}$	5.21$\;\times \;10^{0}$
$f_{10}$	−4.88$\;\times \;10^{2}$ ≈	1.88$\;\times \;10^{0}$	−4.88$\;\times \;10^{2}$ ≈	2.23$\;\times \;10^{0}$	**−4.90$\;\times \;10^{2}$** ≈	**2.42$\;\times \;10^{0}$**	−4.89$\;\times \;10^{2}$	2.29$\;\times \;10^{0}$
$f_{11}$	5.33$\;\times \;10^{2}$ +	1.51$\;\times \;10^{2}$	4.72$\;\times \;10^{2}$ +	1.39$\;\times \;10^{2}$	−3.65$\;\times \;10^{2}$ ≈	8.90$\;\times \;10^{0}$	**−3.66$\;\times \;10^{2}$**	**7.80$\;\times \;10^{0}$**
$f_{12}$	5.69$\;\times \;10^{2}$ +	1.55$\;\times \;10^{2}$	5.84$\;\times \;10^{2}$ +	1.58$\;\times \;10^{2}$	**3.41$\;\times \;10^{2}$** ≈	**9.74$\;\times \;10^{1}$**	4.08$\;\times \;10^{2}$	1.30$\;\times \;10^{2}$
$f_{13}$	7.81$\;\times \;10^{2}$ +	1.59$\;\times \;10^{2}$	7.61$\;\times \;10^{2}$ +	1.48$\;\times \;10^{2}$	5.85$\;\times \;10^{2}$ ≈	1.52$\;\times \;10^{2}$	**5.80$\;\times \;10^{2}$**	**1.35$\;\times \;10^{2}$**
$f_{14}$	8.28$\;\times \;10^{3}$ +	9.40$\;\times \;10^{2}$	8.31$\;\times \;10^{3}$ +	8.13$\;\times \;10^{2}$	**3.21$\;\times \;10^{2}$** ≈	**1.68$\;\times \;10^{2}$**	3.27$\;\times \;10^{2}$	1.62$\;\times \;10^{2}$
$f_{15}$	8.77$\;\times \;10^{3}$ ≈	7.62$\;\times \;10^{2}$	8.90$\;\times \;10^{3}$ ≈	1.04$\;\times \;10^{3}$	**8.49$\;\times \;10^{3}$** ≈	**1.06$\;\times \;10^{3}$**	8.67$\;\times \;10^{3}$	8.16$\;\times \;10^{2}$
$f_{16}$	2.03$\;\times \;10^{2}$ ≈	5.45$\;\times \;10^{-1}$	2.03$\;\times \;10^{2}$ ≈	4.64$\;\times \;10^{-1}$	**2.02$\;\times \;10^{2}$** ≈	**5.70$\;\times \;10^{-1}$**	2.02$\;\times \;10^{2}$	6.19$\;\times \;10^{-1}$
$f_{17}$	2.12$\;\times \;10^{3}$ +	3.14$\;\times \;10^{2}$	2.06$\;\times \;10^{3}$ +	2.77$\;\times \;10^{2}$	**4.08$\;\times \;10^{2}$** ≈	**9.63$\;\times \;10^{0}$**	4.09$\;\times \;10^{2}$	9.50$\;\times \;10^{0}$
$f_{18}$	3.45$\;\times \;10^{3}$ +	5.13$\;\times \;10^{2}$	3.32$\;\times \;10^{3}$ +	4.18$\;\times \;10^{2}$	1.10$\;\times \;10^{3}$ ≈	1.41$\;\times \;10^{2}$	**1.02$\;\times \;10^{3}$**	**1.25$\;\times \;10^{2}$**
$f_{19}$	5.68$\;\times \;10^{2}$ +	1.10$\;\times \;10^{1}$	5.71$\;\times \;10^{2}$ +	1.00$\;\times \;10^{1}$	5.41$\;\times \;10^{2}$ ≈	6.26$\;\times \;10^{0}$	**5.39$\;\times \;10^{2}$**	**6.63$\;\times \;10^{0}$**
$f_{20}$	6.24$\;\times \;10^{2}$ ≈	3.66$\;\times \;10^{-1}$	6.24$\;\times \;10^{2}$ ≈	2.83$\;\times \;10^{-1}$	6.24$\;\times \;10^{2}$ ≈	5.92$\;\times \;10^{-1}$	**6.24$\;\times \;10^{2}$**	**5.46$\;\times \;10^{-1}$**
$f_{21}$	1.19$\;\times \;10^{3}$ +	2.17$\;\times \;10^{2}$	1.15$\;\times \;10^{3}$ +	1.31$\;\times \;10^{0}$	**1.13$\;\times \;10^{3}$** ≈	**3.75$\;\times \;10^{0}$**	1.14$\;\times \;10^{3}$	3.17$\;\times \;10^{0}$
$f_{22}$	1.12$\;\times \;10^{4}$ +	1.17$\;\times \;10^{3}$	1.10$\;\times \;10^{4}$ +	1.23$\;\times \;10^{3}$	1.32$\;\times \;10^{3}$ ≈	1.77$\;\times \;10^{2}$	**1.31$\;\times \;10^{3}$**	**1.33$\;\times \;10^{2}$**
$f_{23}$	1.11$\;\times \;10^{4}$ ≈	1.06$\;\times \;10^{3}$	1.12$\;\times \;10^{4}$ ≈	9.20$\;\times \;10^{2}$	**1.07$\;\times \;10^{4}$** ≈	**1.12$\;\times \;10^{3}$**	1.09$\;\times \;10^{4}$	9.74$\;\times \;10^{2}$
$f_{24}$	1.41$\;\times \;10^{3}$ +	1.29$\;\times \;10^{1}$	1.42$\;\times \;10^{3}$ +	2.55$\;\times \;10^{1}$	**1.36$\;\times \;10^{3}$** ≈	**1.79$\;\times \;10^{1}$**	1.36$\;\times \;10^{3}$	1.47$\;\times \;10^{1}$
$f_{25}$	1.58$\;\times \;10^{3}$ +	2.38$\;\times \;10^{1}$	1.58$\;\times \;10^{3}$ +	2.92$\;\times \;10^{1}$	1.51$\;\times \;10^{3}$ ≈	2.02$\;\times \;10^{1}$	**1.50$\;\times \;10^{3}$**	**1.84$\;\times \;10^{1}$**
$f_{26}$	1.67$\;\times \;10^{3}$ +	1.30$\;\times \;10^{1}$	1.67$\;\times \;10^{3}$ +	1.45$\;\times \;10^{1}$	**1.64$\;\times \;10^{3}$** ≈	**4.53$\;\times \;10^{1}$**	1.64$\;\times \;10^{3}$	1.10$\;\times \;10^{1}$
$f_{27}$	3.57$\;\times \;10^{3}$ +	2.51$\;\times \;10^{2}$	3.60$\;\times \;10^{3}$ +	2.49$\;\times \;10^{2}$	3.25$\;\times \;10^{3}$ ≈	1.59$\;\times \;10^{2}$	**3.25$\;\times \;10^{3}$**	**1.28$\;\times \;10^{2}$**
$f_{28}$	8.88$\;\times \;10^{3}$ +	1.16$\;\times \;10^{3}$	8.96$\;\times \;10^{3}$ +	1.81$\;\times \;10^{3}$	**4.65$\;\times \;10^{3}$** ≈	**1.83$\;\times \;10^{3}$**	4.93$\;\times \;10^{3}$	2.08$\;\times \;10^{3}$
	+/≈/−: 19/7/2		+/≈/−: 19/7/2		+/≈/−: 0/28/0			

## Data Availability

The code of this research can be found at https://github.com/RuiZhong961230/IVEGE.

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
