# Peer review of "Vegetation Evolution with Dynamic Maturity Strategy and Diverse Mutation Strategy for Solving Optimization Problems"

_biomimetics, 2023, doi:10.3390/biomimetics8060454_

Round 1

Reviewer 1 Report

In the manuscript “Vegetation Evolution with Dynamic Maturity Strategy and Diverse Mutation Strategy for Solving Optimization Problems” the authors introduced introduce two new search strategies to further improve the search efficiency of the VEGE and propose an improved VEGE to better balance search efficiency and population diversity. This manuscript is well organized, and the drawn conclusions are coherent with the obtained results. 

Lines 17 - 18: Please, arrange the keywords alphabetically.

Lines 44 – 57: The authors should better describe their hypothesis and predictions.

Lines 23 - 24: I think that you should add these more recent and interesting references to support your sentence: “With their popularity in the field of optimization, a large number of new EC algorithms have been proposed.”. I would like to suggest:

Di Pasquale, G., et al. (2020). Coastal pine-oak glacial refugia in the Mediterranean basin: A biogeographic approach based on charcoal analysis and spatial modelling. Forests, 11(6), 673.

Rahmani, A. M., & AliAbdi, I. (2022). Plant competition optimization: A novel metaheuristic algorithm. Expert Systems, 39(6), e12956.

Rivera-Lopez, R., et al., (2022). Induction of decision trees as classification models through metaheuristics. Swarm and Evolutionary Computation, 69, 101006.

Buonincontri, M. P., et al., (2023). Shedding light on the effects of climate and anthropogenic pressures on the disappearance of Fagus sylvatica in the Italian lowlands: evidence from archaeo-anthracology and spatial analyses. Science of The Total Environment, 877, 162893.

Extensive editing of English language required

Reviewer 2 Report

I am pleased to submit my contributions regarding the revision of the article titled "biomimetics-2539639." This research paper introduces two novel search strategies aimed at improving the performance of the conventional vegetation evolution (VEGE) method. The first strategy, known as the dynamic maturity strategy, allocates more seeds to individuals with better fitness, while the second strategy, called the diverse mutation strategy, increases the diversity of seed individuals through various mutation methods. The study evaluates the performance of these two proposed strategies, comparing them to the conventional VEGE and other advanced evolutionary algorithms (EAs) on benchmark functions and engineering problems. The authors also analyze the contributions of these strategies to the conventional VEGE and confirm that they significantly accelerate convergence speed and improve convergence accuracy in most optimization problems.

The following comments and suggestions are provided to enhance the paper:

Abstract:

Add more details about the methods, main results, values, and general conclusions to make the abstract more robust and informative. Justify the importance of testing the estimation methods more explicitly to show the broader relevance of the study. Additionally, clearly state the objectives of the work to provide readers with a better understanding of its significance.

Introduction:

Include more illustrations of the state of the art in the analysis and methods used to estimate Vegetation Evolution with the Dynamic Maturity Strategy and Diverse Mutation Strategy. Provide a clearer justification for the importance of testing these estimation methods. Clearly describe the objectives of the study to give readers a comprehensive understanding of its purpose.

Material and Methods:

Adhere to the journal's guidelines (https://www.mdpi.com/journal/biomimetics/instructions) and incorporate the sections "2. Vegetation Evolution" and "3. Two Proposed Strategies" into the Material and Methods section. Create a flowchart detailing all the procedures performed in the paper to improve communication with readers. Provide thorough descriptions of each step in the flowchart within the Material and Methods section.

Experimental Evaluations:

Follow the journal's guidelines (https://www.mdpi.com/journal/biomimetics/instructions) and include the Experimental Evaluations section as a subsection of the Results section. Provide more detailed results derived from the methodological framework to improve the quality of the paper.

Discussion:

Include citations supporting the results found in this section and compare the proposed methods and analyses with existing literature results. Consider revising this section entirely, as some topics appear more as results rather than discussion.

Tables and Figures:

Remove Tables 4, 5, 6, 7, 8, 9, and 10, as well as Figures 2, 3, and 4 from the Discussion section. Relocate the figures to the newly created "Results" section and consider placing the tables as supplementary material.

Conclusions:

Add more information about future advances in this field of research based on the results found in this article to strengthen the paper's conclusions.

General:

Provide more detailed captions for figures and tables to enhance the paper's overall quality.

Reviewer 3 Report

The submitted manuscript is titled “Vegetation Evolution with Dynamic Maturity Strategy and Diverse Mutation Strategy for Solving Optimization Problems”, and is concerned with the development of new strategies aiming to improve the performance of the conventional vegetation evolution. In order to achieve the aims of the study, the dynamic mutation strategy and the diverse mutation strategy were introduced The first strategy uses the competitive relationship to rationally allocate resources and expect to generate potential individuals, while the second strategy provides a variety of mutation methods to increase diversity and the ability to escape from local areas. Benchmark tests and comparisons with other algorithms have been carried out to verify the effectiveness of the strategies. However, there are some serious shortcomings that need to be addressed to prove the effectiveness of the new strategies:

(1)   Differential evolution (DE) is among the algorithms to compare. Table 3 contains the settings of DE: DE/current-to-rand/1/bin, F=1, Cr=0.9. As far as I understand, these parameters are used for unimodal and multimodal functions. It is well known and it is confirmed in many studies that each objective function requires optimal settings. Figure 2 shows the poor convergence of DE for different functions, but in fact it could be (and I am sure it is) that the chosen strategy as well as F and Cr are not optimal for the functions to be minimized. And such a comparison is illegitimate. Most likely, the same is true for other algorithms.  

It seems that the authors run the other optimization algorithms with not optimal parameters trying to prove that their improvement provides the best convergence. In general, the authors must demonstrate that competing algorithms run at the top of their performance.

(2)   The bibliography looks poor and requires additional related references.

The manuscript can be published after major revision.

Reviewer 4 Report

1. In the introduction there are no references to the article by Mehrabian A.R., Lucas S. on the Weed Colonization method.

2. The second term in (2) should be written in matrix form.

3. When describing equation (3), there must be a reference to the DE method.

4.After expression (4), the number of seeds generated by each individual should be mathematically described.

5. Probability takes values from 0 to 1. Lines 139 and 147 contain erroneous values.

6. The binary crossover operation must be described accordingly (line 143).

Round 2

Reviewer 2 Report

We thank the authors for accepting the suggestions and making the suggested modifications. The current version of the paper biomimetics-2539639 is ready to be accepted in its current form.

Author Response

We would appreciate your effort for improving the quality of our paper.

Reviewer 3 Report

Thanks for your reply. I agree with the revisions.

Fine.

Author Response

We would appreciate your effort in improving the quality of our paper.

Reviewer 4 Report

1. In formulas (2) and (3) it is recommended to use the product of vectors according to Hadamard. Otherwise, coordination of dimensions is required for the existence of the product.

2. In lines 153 and 161 there are expressions for probability in percentage, which does not correspond to the definition of probability and is therefore erroneous. The reviewer's comment remained uncorrected.
